# Sequence and parent-of-origin dependent m⁶A contribute to allele-specific gene expression

Ying Zhang [1,2,3,9], Ze-Yu Zhang [4,9], Hong-Xuan Chen[1,5,6,9], Chao Liu[7], Biao-Di Liu[1], Ye-Lin Lan[1], Ying-Yuan Xie[1,5,6], Tao Chen[1], Shaobo Chen[4], Guihai Feng[7], Zhang Zhang[1], Wei Li [7], Nan Cao[2], Xiu-Jie Wang [4] & Guan-Zheng Luo [1,5,6,8]✉

## Abstract

**Multiple regulatory layers influence allele-specific expression (ASE), particularly through sequence-dependent and parent-of-origin-dependent mechanisms at the transcriptional level. However, little is known about ASE regulation at the post-transcriptional level. The most prevalent post-transcriptional mRNA modification, $N^6$-methyladenosine (m⁶A), plays important roles in regulating gene expression. Here, we conduct transcriptome-wide analysis of allele-specific m⁶A in mice. Using early postnatal tissues from reciprocal crosses of two divergent mouse strains, we measured allelic m⁶A differences at single-base resolution. Our study reveals widespread sequence-dependent allelic imbalance in m⁶A methylation, identifying thousands of allele-specific m⁶A (ASm⁶A) sites with statistically significant and reproducible allelic methylation differences. We find evidence of potential *cis*-regulatory variants within 50-nt flanking regions of ASm⁶As. Intriguingly, we detect parental effects on allelic methylation across m⁶As exhibiting parent-of-origin-dependent ASE. For both sequence- and parent-of-origin-dependent m⁶As, we observe opposing allelic preferences between methylation and expression, suggesting a potential role of ASm⁶A in regulating ASE through negative effects on gene expression. Overall, our findings reveal that both *cis*-acting and parent-of-origin effects influence ASm⁶A, offering new insights into post-transcriptional mechanisms of ASE regulation.**

**Keywords** m⁶A Modification; Allele-specific Expression; Cis Regulatory Element; Parent-of-origin Effect; Genomic Imprinting
**Subject Category** RNA Biology

## Introduction

The study of allele-specific events in diploid organisms has provided a unique perspective to understand the regulatory mechanisms governing gene expression. Genetic or epigenetic variations between alleles can influence allele-specific expression (ASE) through a range of allele-specific events (Cleary and Seoighe, 2021; Onuchic et al, 2018), encompassing chromatin structure (Noordermeer and Feil, 2020; Richer et al, 2023; Zhang et al, 2020a), transcription factor (TF) binding (Benaglio et al, 2019; Lleres et al, 2019), DNA methylation (Onuchic et al, 2018; Xie et al, 2012), histone modification (Guo et al, 2015; Inoue et al, 2017; Onuchic et al, 2018; Sungalee et al, 2021), mRNA splicing (Amoah et al, 2021; Nembaware et al, 2008), and RNA-binding protein (RBP) binding (Bahrami-Samani and Xing, 2019; Yang et al, 2019). Allele-specific analyses in mammals have revealed widespread sequence-dependent ASE, primarily attributed to *cis*-acting effects that affect allelic imbalances in diverse transcriptional regulatory processes (Bryois et al, 2014; Crowley et al, 2015; Benaglio et al, 2019; Lleres et al, 2019; Onuchic et al, 2018; Xie et al, 2012). Another representative type of ASE in mammals is parent-of-origin-dependent ASE, with exclusive or preferential expression favoring the allele of paternal or maternal origin (Delaval and Feil, 2004; Perez et al, 2015; Tucci et al, 2019). Genes exhibiting such ASE are usually termed imprinted genes, constituting over 200 loci within the mammalian genome (Crowley et al, 2015; Gregg et al, 2010; Perez et al, 2015; Tucci et al, 2019). Allele-specific investigations have demonstrated that specific epigenetic marks, such as DNA methylation and histone modifications, have the capability to retain parental origin information and regulate the expression of imprinted genes (Barlow, 2011; Delaval and Feil, 2004; Inoue et al, 2017; Tucci et al, 2019; Zink et al, 2018). Nonetheless, whether additional mechanisms, particularly at the post-transcriptional level, contribute to the control of sequence-dependent or parent-of-origin-dependent ASE remains an open question. The exploration of novel allele-specific events and their association with ASE holds potential for unveiling new gene

[1]MOE Key Laboratory of Gene Function and Regulation, Guangdong Province Key Laboratory of Pharmaceutical Functional Genes, State Key Laboratory of Biocontrol, School of Life Sciences, Sun Yat-sen University, 510275 Guangzhou, China. [2]Zhongshan School of Medicine, Sun Yat-sen University, 510080 Guangzhou, China. [3]The Center for Ion Beam Bioengineering & Green Agriculture, Hefei Institutes of Physical Science, Chinese Academy of Sciences, 230031 Hefei, China. [4]Key Laboratory of Genetic Network Biology, Institute of Genetics and Developmental Biology, Chinese Academy of Sciences, 100101 Beijing, China. [5]Innovation Center for Evolutionary Synthetic Biology, Sun Yat-sen University, 510275 Guangzhou, China. [6]Sun Yat-sen University Institute of Advanced Studies Hong Kong, Science Park, 999077 Hong Kong SAR, China. [7]State Key Laboratory of Stem Cell and Reproductive Biology, Institute of Zoology, Chinese Academy of Sciences, 100101 Beijing, China. [8]Pingshan Translational Medicine Center, Shenzhen Bay Laboratory, 518118 Shenzhen, China. [9]These authors contributed equally: Ying Zhang, Ze-Yu Zhang, Hong-Xuan Chen. ✉E-mail: luogzh5@mail.sysu.edu.cn

expression mechanisms and shedding light on the biological significance of ASE.

Post-transcriptional gene regulation, particularly through RNA modifications, is increasingly recognized as a crucial layer of gene expression control (Frye et al, 2018; Ontiveros et al, 2019; Roundtree et al, 2017). Among the diverse RNA modifications, $N^6$-methyladenosine ($m^6A$) stands out as the most abundant mRNA modification and plays a central role in regulating mRNA fate (Frye et al, 2018; Gilbert et al, 2016; Murakami and Jaffrey, 2022; Ontiveros et al, 2019; Roundtree et al, 2017). Installed within a consensus DRACH motif (where D = G/A/T, R = A/G, H = A/C/T, and A represents the methylatable adenosine) by the METTL3/METTL14 methyltransferase complex (He and He, 2021; Liu et al, 2014; Murakami and Jaffrey, 2022; Wang et al, 2016), $m^6A$ exhibits a marked transcriptomic distribution, primarily enriching in 3′ untranslated regions (3′ UTRs) and stop codon proximal regions (Dominissini et al, 2012; He et al, 2023; Meyer et al, 2012; Murakami and Jaffrey, 2022; Uzonyi et al, 2023; Yang et al, 2022). $m^6A$ regulates gene expression by influencing the entire mRNA life cycle, including pre-mRNA processing, nuclear export, decay, and translation (Frye et al, 2018; He and He, 2021; Murakami and Jaffrey, 2022; Roundtree et al, 2017). These processes are mainly mediated by $m^6A$ "reader" proteins that selectively recognize $m^6A$ and exert regulatory functions on the $m^6A$-marked mRNA (Shi et al, 2019; Yang et al, 2018; Zaccara et al, 2019). Notably, the major effect of $m^6A$ on mRNAs is promoting mRNA degradation in the cytoplasm (Lee et al, 2020; Murakami and Jaffrey, 2022; Shi et al, 2017; Zaccara and Jaffrey, 2020), as initially described in seminal work in 1978 (Sommer et al, 1978; Meyer, 2022). Furthermore, the critical involvement of $m^6A$ methylation in various physiological and pathophysiological processes underscores its immense biological significance as a post-transcriptional gene expression regulator (Frye et al, 2018; He and He, 2021; Murakami and Jaffrey, 2022; Roundtree et al, 2017; Zhang et al, 2020b).

Given the pivotal role of $m^6A$ in mRNA fate, studying allele-specific $m^6A$ (ASm$^6A$) offers a unique perspective for investigating ASE regulation in the post-transcriptional level. While recent studies have begun to identify ASm$^6A$s and associated genetic variants (Cao et al, 2023; Xu et al, 2021), the correlation between ASm$^6A$ and ASE remains largely unknown. Quantitative trait locus (QTL) studies have linked multiple genetic loci to $m^6A$ levels in human tissues (Xiong et al, 2021) and Yoruba lymphoblastoid cell lines (Zhang et al, 2020b). Interestingly, a small subset of these loci colocalize with expression QTLs (eQTLs) which are predominantly associated with mRNA degradation (Xiong et al, 2021), suggesting a potential connection between ASm$^6A$ and ASE. Furthermore, both $m^6A$ QTLs and the regulatory variants for ASm$^6A$ colocalize with many disease-associated loci (Cao et al, 2023; Xiong et al, 2021; Zhang et al, 2020b). Despite the importance of these regulatory loci, their positioning relative to the regulated $m^6A$ sites remains unclear. This gap is attributed to the peak-level quantification of $m^6A$ in these studies (Cao et al, 2023, Xiong et al, 2021, Zhang et al, 2020b), hindering precise $m^6A$ localization and identification of nearby cis-regulatory elements. Furthermore, the influence of parent-of-origin effects on ASm$^6A$s, particularly at imprinted loci, remains completely uncharacterized.

Here, we present the first transcriptome-wide analysis of allele-specific $m^6A$ methylation in mice, utilizing cerebellum and cerebrum samples from reciprocal crosses of two divergent strains

at postnatal day 0 (P0) and day 7 (P7). The selection of these two tissue types at early postnatal stages was guided by the significant role of both $m^6A$ and ASE in brain development and function (Livneh et al, 2020; Kravitz and Gregg, 2019), especially in the context of parent-of-origin effects (Perez et al, 2015; Gregg et al, 2010; Huang et al, 2018; Tucci et al, 2019). Our findings reveal pervasive sequence-dependent allelic imbalance in $m^6A$ methylation, with 4692 identified ASm$^6A$ sites showing significant allelic bias. The direction and level of allelic bias at these sites exhibit high reproducibility across diverse samples, indicating consistency of underlying cis-regulatory effects. Potential cis-regulatory loci affecting allelic $m^6A$ levels are enriched within the 50-nt flanking regions of $m^6A$, with the most significant enrichment occurring at the motif positions. Notably, our study provides evidence for parental effects on $m^6A$ methylation in genes exhibiting parent-of-origin-dependent ASE. Interestingly, for both sequence-dependent and parent-of-origin-dependent $m^6A$ sites, we observed contrasting allelic preferences between methylation and expression, suggesting a potential role of ASm$^6A$ in regulating ASE through negative effects on gene expression. Collectively, our findings indicate that ASm$^6A$ is influenced by both cis-acting and parent-of-origin effects, potentially regulating ASE at the post-transcriptional level.

# Results

## Allele-specific $m^6A$ profiles reveal pervasive sequence-dependent allelic imbalance

To identify allele-specific $m^6A$ methylation, we carried out reciprocal crosses between two mouse strains with distant genetic backgrounds (PWK/PhJ (PWK) and C57BL/6J (C57)) and conducted multiplexed $m^6A$-seq (Dominissini et al, 2012; Meyer et al, 2012; Dierks et al, 2021) and GLORI (glyoxal and nitrite-mediated deamination of unmethylated adenosines) (Liu et al, 2023) assays on early postnatal cerebellum and cerebrum tissues from F1 hybrids (Figs. 1A and EV1A; Table 1 and Table EV1). These hybrids were generated through both initial (F1i; PWK mother × C57 father) and reciprocal (F1r; C57 mother × PWK father) crosses. To define high-confidence single-base $m^6A$ sites for the following allelic methylation analysis, we first overlapped $m^6A$ peaks identified by $m^6A$-seq with GLORI-detected $m^6A$s in the same tissue (see "Methods"). These $m^6A$ sites exhibited the expected enrichment pattern in 3′ UTRs and near stop codons (Fig. EV1B–D), along with the enrichment for the canonical DRACH motif (Fig. EV1D). Additionally, a high degree of reproducibility was observed, with >91% of these sites detected in more than three samples (Fig. EV1E). For downstream differential allelic methylation analysis, we defined allelically detectable $m^6A$ sites that meet stringent allelic coverage criteria (see "Methods"). The distribution of these sites along mRNA transcripts exhibited a slight shift towards the 3′ UTR, likely reflecting the increased density of single-nucleotide polymorphism (SNP) sites in this region (Figs. 1B and EV1F), which is necessary for allele-specific quantification. In total, we identified 74,753 high-confidence $m^6A$ sites, of which ~32% ($n = 23{,}654$) showed detectable allelic levels in at least one sample for further allele-specific $m^6A$ exploration.

We next quantified allelic $m^6A$ levels based on allele-specific alignments in each $m^6A$-seq sample, enabling us to split the $m^6A$

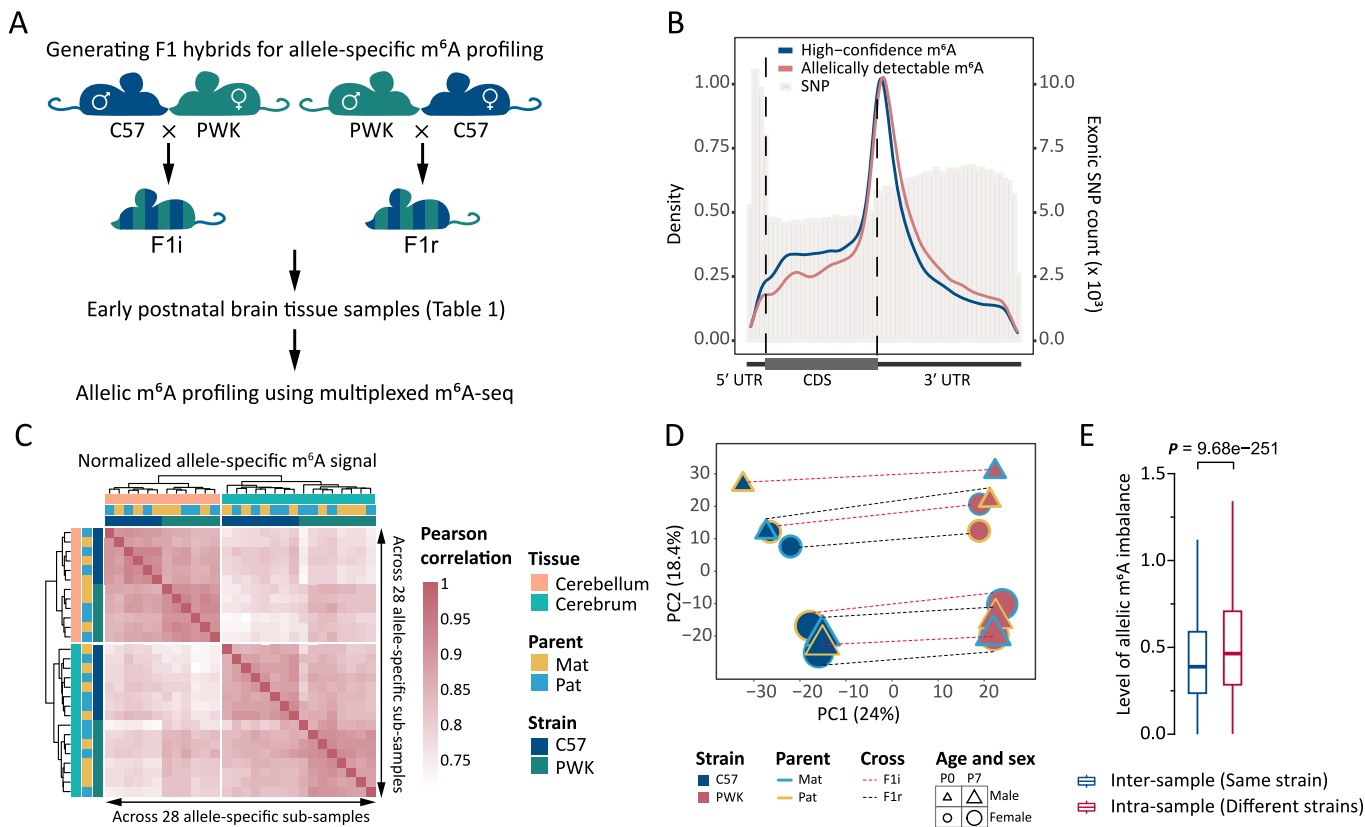

**Figure 1. Study design and allele-specific m6A profiles in hybrid mice.**

(A) Overview of sample collection and methods for transcriptome-wide profiling of allele-specific m6A. Sample details are provided in Table 1. (B) Metagene profiles showing the distributions of high-confidence m6A sites ($n = 70{,}709$), allelically detectable m6A sites ($n = 22{,}969$) and exonic SNPs. (C) Pearson correlation (heatmap) and hierarchical clustering of allele-specific m6A profiles across 28 sub-samples, revealing tissue- and strain-specific clustering ($n = 2153$ allelically detectable m6A sites shared across all samples). (D) PCA of allele-specific m6A levels in 16 cerebrum sub-samples. Each dot represents an allele-specific sub-sample, with shape denoting sex (circle for female, triangle for male) and size denoting age (small for P0, large for P7). Dot color indicates genotype, with outline color denoting parent-of-origin and fill color denoting strain. Dotted lines connect the two sub-samples from each physical sample, with line color indicating cross (F1i or F1r). (E) Box plots comparing inter-sample same-strain and intra-sample different-strain allelic m6A imbalance levels in the F1i-F1r group from P0 male cerebellum ($n = 14{,}125$ m6A sites). Other groups followed the same trend. The top, middle, and bottom lines of the box represent the upper quartile (Q3), median, and lower quartile (Q1), respectively. The upper whisker extends to the maximum value provided it is not larger than ($Q3 + 1.5 \times IQR$) (where $IQR = Q3 - Q1$), while the lower whisker extends to the minimum value provided it is not smaller than ($Q1 - 1.5 \times IQR$). Data points beyond the whiskers are considered outliers and are not displayed. Statistical analysis was conducted using the one-sided paired Wilcoxon rank-sum test. Source data are available online for this figure.

profile from each sample into two allele-specific profiles (sub-samples; see "Methods"). Unsupervised clustering of these allele-specific m6A profiles revealed a clear separation based on tissue type, indicating its dominant influence on m6A methylation profiles (Fig. 1C). Within each tissue, the major driver of differential allelic m6A methylation was strain- or sequence-dependent effect, which significantly outweighed the parent-of-origin effect (Fig. 1C). Further analysis using principal component analysis (PCA) on allele-specific m6A levels for each tissue revealed that the first principal component (PC1) explained >24% of the variation and strongly correlated with strain in both tissues (Figs. 1D and EV1G,H). Interestingly, the second principal component (PC2), accounting for over 18% of the variance, was predominantly associated with sex in the cerebellum, while associated with age in the cerebrum (Figs. 1D and EV1G,H). These findings highlight substantial divergence between C57 and PWK allele-specific m6A profiles within each tissue, even when derived from the same sample. This disparity, primarily influenced by strain- or sequence-

dependent effects, may also be affected by other factors, such as sex and age, with their impact varying depending on the tissue type.

To further dissect the impact of strain on allele-specific m6A profiles, we extended our analysis to encompass all allelically detectable m6As in each F1i-F1r group (The two age- and sex-matched F1i and F1r samples are from here on referred to as a group.) (Table 1; see "Methods"). Within-group comparisons of allelic m6A imbalance levels allowed us to investigate the effect of strain while controlling for potential confounding factors like age and sex. Specifically, for each m6A site, two metrics were introduced based on the four allele-specific m6A levels within an F1i-F1r group: the inter-sample same-strain and intra-sample different-strain allelic m6A imbalance levels (see "Methods"). Remarkably, across all F1i-F1r groups, the different-strain metrics consistently exhibited significantly higher levels compared to the same-strain metrics (paired Wilcoxon rank-sum test, $P < 2.2 \times 10^{-16}$; Fig. 1E), highlighting the dominant role of sequence context in shaping allele-specific m6A profiles. This analysis encompassed

**Table 1.  Brain tissue samples used in this study.**

| Sample ID | Tissue | Age | Sex | Cross | Detection method | F1i-F1r group ID |
|---|---|---|---|---|---|---|
| bel_P0_m_F1i | Cerebellum | P0 | Male | F1i | m⁶A-seq | bel_P0_m |
| bel_P0_m_F1r | Cerebellum | P0 | Male | F1r | m⁶A-seq | |
| bel_P7_m_F1i | Cerebellum | P7 | Male | F1i | m⁶A-seq | bel_P7_m |
| bel_P7_m_F1r | Cerebellum | P7 | Male | F1r | m⁶A-seq | |
| bel_P7_f_F1i | Cerebellum | P7 | Female | F1i | m6A-seq | bel_P7_f |
| bel_P7_f_F1r | Cerebellum | P7 | Female | F1r | m⁶A-seq | |
| bru_P0_m_F1i | Cerebrum | P0 | Male | F1i | m⁶A-seq | bru_P0_m |
| bru_P0_m_F1r | Cerebrum | P0 | Male | F1r | m⁶A-seq | |
| bru_P0_f_F1i | Cerebrum | P0 | Female | F1i | m⁶A-seq | bru_P0_f |
| bru_P0_f_F1r | Cerebrum | P0 | Female | F1r | m⁶A-seq | |
| bru_P7_m_F1i | Cerebrum | P7 | Male | F1i | m⁶A-seq | bru_P7_m |
| bru_P7_m_F1r | Cerebrum | P7 | Male | F1r | m⁶A-seq | |
| bru_P7_f_F1i | Cerebrum | P7 | Female | F1i | m⁶A-seq | bru_P7_f |
| bru_P7_f_F1r | Cerebrum | P7 | Female | F1r | m⁶A-seq | |
| bel_P0_f_F1r_G | Cerebellum | P0 | Female | F1r_G | GLORI | / |
| bru_P0_f_F1r_G | Cerebrum | P0 | Female | F1r_G | GLORI | |

~8000–30,000 m⁶A sites within each F1i-F1r group, revealing a notable prevalence of sequence-dependent effects on allelic m⁶A imbalance. These findings collectively indicate the potential for widespread allele-specific m⁶A, or ASm⁶A, throughout the epitranscriptome of F1 hybrid mice.

## Identification of allele-specific m⁶As

To rigorously identify ASm⁶A sites, we employed a binomial model to statistically assess allelic methylation differences at each m⁶A site using the allele-specific alignments from m⁶A-seq datasets (see "Methods"). For enhanced statistical power, we selectively chose m⁶A sites with high allelic coverage (testable m⁶A sites) in each F1i-F1r group for downstream ASm⁶A identification (see "Methods"; Fig. EV2A). We quantified allelic m⁶A imbalance levels using the fold change (FC) of m⁶A methylation between two alleles. Two metrics, $\log_2(cpFC)$ and $\log_2(mpFC)$, were introduced to measure sequence-dependent (C57/PWK) and parent-of-origin-dependent (maternal/paternal) allelic differences, respectively (see "Methods"). To establish an optimal threshold for ASm⁶A identification and evaluate stochastic fluctuations in $\log_2(cpFC)$ values, we utilized testable m⁶As at SNP positions (i.e., strain-specific adenine sites) as internal positive controls. These control sites are strain-specific adenine bases, ensuring predictable allelic bias. We calculated Euclidean distances between the medians of positive and negative $\log_2(cpFC)$ values to evaluate their stochastic fluctuations (see "Methods"; Fig. EV2B,C). Based on this analysis, we set the threshold of absolute $\log_2(cpFC)$ to 0.6 for ASm⁶A identification. Notably, ~80% of positive controls exhibited the expected allelic m⁶A preference (C57-biased or PWK-biased based on the specific presence of adenine base in one strain) (Figs. 2A and EV2D; Dataset EV1).

Following the analyses described above, we employed specific criteria ($P < 0.05$, FDR $< 0.1$ and $|\log_2(cpFC)| > 0.6$) to identify candidate ASm⁶As in each m⁶A-seq sample (see "Methods"). This approach was further refined by requiring presence in both F1i and F1r samples within each F1i-F1r group (Fig. 2B). We then categorized the identified ASm⁶A sites into two major classes: sequence-dependent ASm⁶As (seq-ASm⁶As) and parent-of-origin dependent ASm⁶As (parent-ASm⁶As) (see "Methods"; Figs. 2B and EV2E). Notably, the vast majority of ASm⁶A sites displayed sequence-dependent allelic methylation biases, with only sporadic instances of parent-of-origin-dependent allelic m⁶A imbalance. This distribution pattern is consistent with previous findings on ASE (Crowley et al, 2015) and allele-specific DNA methylation (ASM) (Xie et al, 2012), suggesting potential analogous regulatory mechanisms governing these diverse allele-specific events.

We then identified highly reproducible ASm⁶As based on their reproducibility in allelic bias direction (see "Methods"; Fig. EV2F). Notably, ~86% of seq-ASm⁶A sites exhibit high reproducibility, demonstrating consistent allelic bias directions across F1i-F1r groups in each tissue (Fig. 2C). This finding indicates that seq-ASm⁶As are governed by relatively stable regulatory mechanisms, rendering them less susceptible to sex and age influences in the same tissue. In contrast, only a minor fraction of parent-ASm⁶As demonstrated reproducibility (Fig. 2C), and they generally exhibited mild allelic imbalance levels (Fig. EV2F). This suggests that parental effects on m⁶A may be subtle and potentially masked or influenced by other factors such as sex, age, or specific sequence contexts.

Overall, our analysis identified 2383 (1859 highly reproducible) ASm⁶As in the cerebellum and 2309 (1942 highly reproducible) ASm⁶As in the cerebrum, with 350 highly reproducible seq-ASm⁶A sites (17%) shared across both tissues (Dataset EV2). Notably, the tissue-shared seq-ASm⁶As displayed 100% directional consistency in allelic bias and demonstrated a strong correlation in allelic imbalance levels (Fig. 2D), providing robust confirmation of the identified seq-ASm⁶A sites. These findings collectively underscore the prevalence and consistent directionality of sequence-dependent or *cis*-regulatory effects on m⁶A levels across diverse samples.

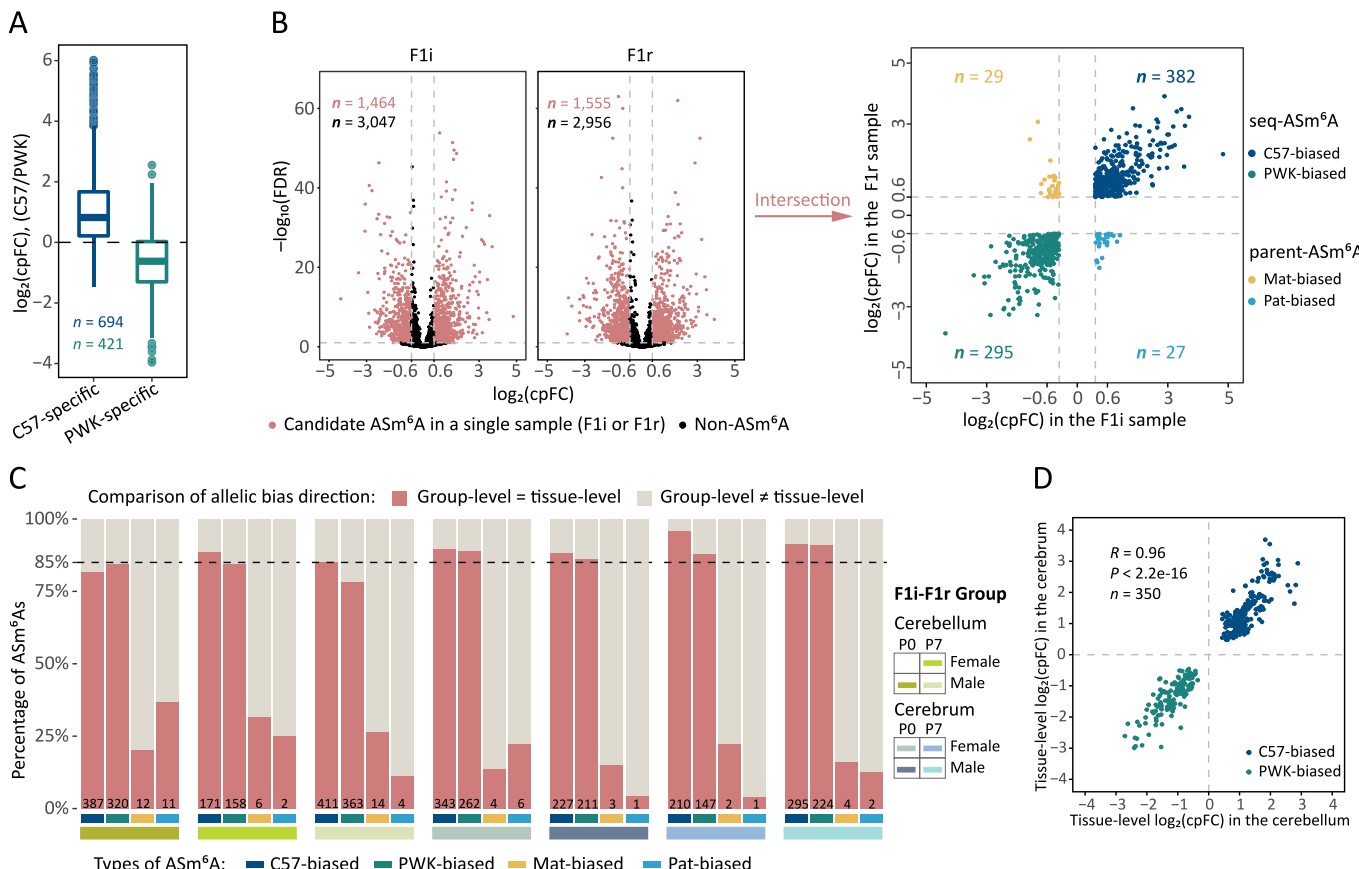

**Figure 2. Characterization of ASm6A sites.**

(A) Box plot illustrating the distribution of $\log_2(cpFC)$ values for positive controls, including C57-specific and PWK-specific m6A sites. The top, middle, and bottom lines of the box represent the upper quartile (Q3), median, and lower quartile (Q1), respectively. The upper whisker extends to the maximum value provided it is not larger than $(Q3 + 1.5 \times IQR)$ (where $IQR = Q3 - Q1$), while the lower whisker extends to the minimum value provided it is not smaller than $(Q1 - 1.5 \times IQR)$. Data points beyond the whiskers are considered outliers and are plotted individually. (B) Scatter plots depict the ASm6A identification process within an F1i-F1r group (the P0 female cerebrum group). Volcano plots illustrate the identification of candidate ASm6A sites in individual F1i and F1r samples. The intersection of these candidates defines ASm6A sites for the group, further categorized based on their allelic bias directions in F1i and F1r samples. Each point represents an m6A site, with color indicating the ASm6A type. The number of m6A sites ($n$) for each type is marked in the same color within the figure. (C) Stacked bar charts illustrating the proportion of ASm6As demonstrating high reproducibility. The dashed line denotes the mean proportion for seq-ASm6A sites. (D) Scatter plot showing the correlation of allelic m6A imbalances between cerebellum and cerebrum, utilizing shared seq-ASm6A sites in both tissues. Allelic m6A bias directions at these sites exhibited complete consistency in both tissues. Pearson's correlation analysis was performed to evaluate the linear relationship ($R = 0.96$, $P = 4.25e-204$). Source data are available online for this figure.

## Cis-regulatory effects on seq-ASm6As

We next asked whether the tissue-specific seq-ASm6As, constituting ~83% of all identified seq-ASm6As, also show high level of consistency in allelic preferences among F1i-F1r groups. To address this, we compared group-sharing patterns in seq-ASm6As and testable m6A sites. Seq-ASm6A sites showed a group-sharing rate of ~41%, significantly lower than the ~69% observed for testable m6A sites (Fig. EV3A). Our analysis indicated that this difference was a consequence of the stringent criteria used to select seq-ASm6As, not inter-group differences in allelic preference. We evaluated whether group-specific seq-ASm6As maintain consistent allelic bias directions in other groups where they were testable (but not identified as seq-ASm6A sites). Remarkably, we observed substantial agreements in allelic m6A preferences (>91%) and strong positive correlations ($R > 0.84$, $P < 2.2e-16$) between seq-ASm6A sites in one F1i-F1r group (group 1) and testable sites in another group (group 2), even

across different tissues (Figs. 3A,B and EV3B). These findings suggest that cis-regulatory effects influencing seq-ASm6A sites are largely shared across samples from various tissues, developmental stages, and sexes.

Given the prevalence of sequence-dependent ASm6A in hybrid mice, we next investigated cis-regulatory variants as potential drivers of such allelic methylation imbalance. To this end, we examined the density and distances of SNPs around m6As. We observed a higher SNP density in the neighboring regions of seq-ASm6A sites compared to non-seq-ASm6A sites (Fig. 3C). Furthermore, these seq-ASm6As exhibited significantly shorter distance to their nearest SNPs (Wilcoxon rank-sum test, $P < 2.2e-16$). These observations indicate that local sequence context plays a pivotal role in regulating m6A levels. To further quantify SNP distribution in the flanking 100-nt region of seq-ASm6As, we introduced a weighted scoring metric wherein higher scores denote higher density in closer regions (see "Methods"). Notably, the score

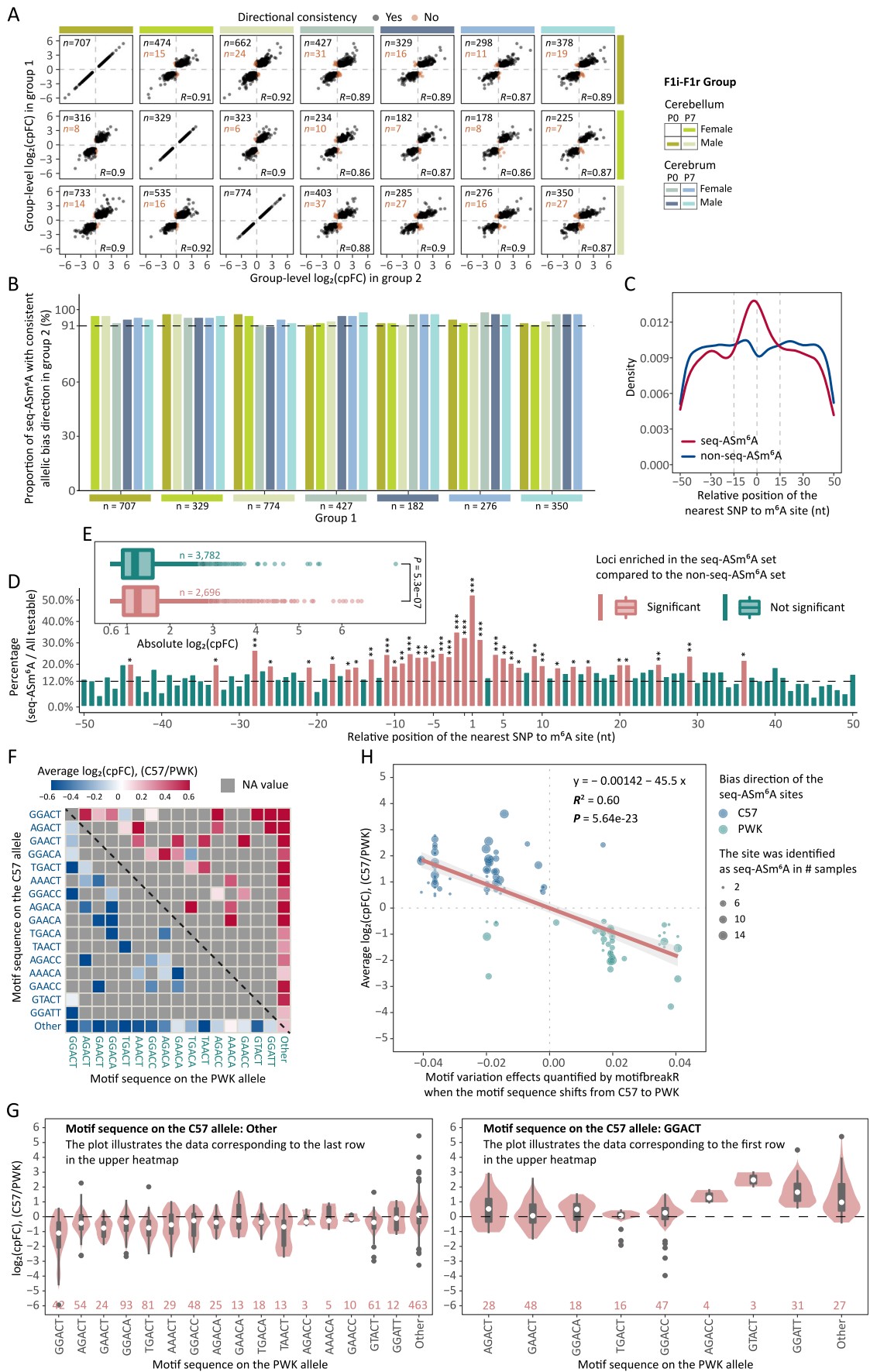

**Figure 3. *Cis*-regulatory effects on seq-ASm⁶A sites.**

(A) Correlation analysis of allelic biases in seq-ASm⁶A methylation across F1i-F1r groups. Each scatterplot depicts the intersection between seq-ASm⁶As in a cerebellar group (Group 1) and allelically detectable m⁶A sites in another group (Group 2). Pearson's *R* and the counts of m⁶A sites are annotated. (B) Bar plots illustrating the proportions of seq-ASm⁶A sites exhibiting consistent allelic bias directions between groups, with the mean proportion indicated by a dashed line. Color scheme for F1i-F1r groups is as in (A). (C) Density plot depicting the relative positions of the nearest SNPs for seq-ASm⁶A and non-seq-ASm⁶A sites. (D) Enrichment analysis of SNP positions in the flanking regions of seq-ASm⁶A sites (see "Methods"). Each bar represents the ratio of seq-ASm⁶As to all testable m⁶A sites with their nearest SNPs at the position. The dashed line indicates the average ratio. Statistical analysis employed the one-sided Binomial test (*$P$ <0.05, **$P$ < 0.01, ***$P$ < 0.0001). The $P$ values for the significant positions ($-44, -33, -28, -26, -21, -18, -16, -15, -13, -11, -10, -9, -8, -7, -6, -5, -4, -3, -2, -1, 1, 2, 4, 5, 6, 7, 9, 10, 12, 14, 16, 20, 21, 25, 29, 36$) are as follows: 0.024, 0.032, 0.0002, 0.02, 0.038, 0.011, 0.038, 0.038, 0.002, 0.00005, 0.014, 0.008, 0.00001, 0.00002, 0.0002, 0.0004, 3.3e-06, 0.00007, 1.7e-12, 2.5e-08, 8.0e-30,1.6e-09, 7.0e-06, 0.0009, 0.005, 0.015, 0.0004, 0.006, 0.039, 0.011, 0.012, 0.015, 0.011, 0.007, 0.002, 0.021. The positions and $P$ values are in corresponding order. (E) Box plot illustrating the distributions of allelic imbalance levels for seq-ASm⁶As with the nearest SNP at the two types of positions depicted in (D). The number of seq-ASm⁶A sites are indicated ($n = 3782$ and $n = 2696$ sites for each type). Statistical analysis utilized the two-sided Wilcoxon test. (F) Heatmap depicting allelic m⁶A differences among motif pairs in the cerebellum (see "Methods"). Motifs are ranked by occurrence frequency within all high-confidence m⁶A sites in the cerebellum. For each motif pair, color represents the average $\log_2(cpFC)$ value of m⁶A sites with corresponding motif variations. Blue and red indicate higher m⁶A levels on the PWK and C57 alleles, respectively. Gray represents NA values. (G) Violin plots detail the distributions of allelic methylation differences at m⁶A sites with specific motif variations. For each motif variation, the count of m⁶A sites are labeled (red numbers along the $x$ axis). White circle indicates median, thick gray vertical line represents IQR, thin vertical lines extend up to 1.5× IQR, and violin-shaped areas depict kernel density estimates of data distribution. (H) Linear regression analysis of motif variation effects and allelic m⁶A differences in cerebellar seq-ASm⁶As. The variation effects were determined using motifbreakR (see "Methods"). Shaded areas represent 95% confidence intervals of the regression line. Statistical significance was assessed using the Student's *t* test. (C, D) SNP positions are labeled with respect to the transcript strand. (E, G) The top, middle, and bottom lines of the box represent the upper quartile (Q3), median, and lower quartile (Q1), respectively. The upper whisker extends to the maximum value provided it is not larger than ($Q3 + 1.5× IQR$) (where $IQR = Q3 - Q1$), while the lower whisker extends to the minimum value provided it is not smaller than ($Q1 - 1.5× IQR$). Data points beyond the whiskers are considered outliers and are plotted individually.

exhibited a positive correlation with the allelic imbalance level of seq-ASm⁶A sites, reaffirming the presence of critical *cis*-regulatory variants adjacent to seq-ASm⁶As (Fig. EV3C). Importantly, >90% of seq-ASm⁶A sites have no variant in the motif region, suggesting widespread *cis*-regulatory effects originating from other flanking loci.

We then predicted candidate regulatory loci governing m⁶A levels based on enrichment analysis of SNP locations in the flanking 100-nt regions of seq-ASm⁶A sites (see "Methods"). Remarkably, all four positions spanning the 5-mer motif ($-2, -1, +1$ and $+2$ relative to the m⁶A) exhibited significant enrichment, with the $+1$ position showing the highest significance (Fig. EV3D). Among all testable sites with SNPs at the $+1$ position, a remarkable 53% were identified as highly reproducible seq-ASm⁶A sites (Fig. EV3D), demonstrating the high susceptibility of m⁶A levels to genetic variations at this specific position. Interestingly, all candidate *cis*-regulatory loci identified through the analysis were found within the flanking 50-nt regions of seq-ASm⁶A sites, with a higher number of loci in upstream regions compared to downstream regions (Fig. 3D). Furthermore, we observed significantly elevated allelic imbalance levels of seq-ASm⁶As with their nearest SNPs located at these candidate *cis*-acting loci (Figs. 3E and EV3E). These findings further support the involvement of specific *cis*-regulatory loci in modulating m⁶A levels, with a notable emphasis on those associated with motif positions.

To explore the influence of motif variations on allelic m⁶A imbalance, we focused our analysis on allelically detectable m⁶A sites with distinct motif sequences between the two alleles (see "Methods"). Intriguingly, we found that the allele harboring the more frequently occurring motif sequence predominantly exhibited a higher methylation level compared to the other allele (Figs. 3F,G and EV3F). Furthermore, we observed strong directional agreement between allelic m⁶A bias and motif variation effects quantified by motifbreaR (Coetzee et al, 2015) (see "Methods"; Figs. 3H and EV3G). These findings indicate that genetic variations at motif positions exert consistent and predictable influences on m⁶A methylation levels. Notably, the extent of such influence is mainly

determined by the relative frequency of the motif sequences. These frequencies may reflect the varying binding affinities between motif sequences and regulators like m⁶A writers, erasers, or RNA-binding proteins (RBPs) responsible for modulating m⁶A methylation.

## Sequence-dependent ASm⁶A methylation negatively associated with ASE

Previous studies of ASE in highly divergent mouse crosses have revealed pervasive *cis*-regulatory variation (Cleary and Seoighe, 2021; Crowley et al, 2015). Given the established crucial role of m⁶A in gene expression regulation (Frye et al, 2018; He and He, 2021; Murakami and Jaffrey, 2022; Roundtree et al, 2017), we asked whether seq-ASm⁶A could influence sequence-dependent ASE. To address this, we calculated paired allelic read ratios (ARRs) for untreated input ($ARR_{c57,input}$) and immunoprecipitated (IP) samples ($ARR_{c57,ip}$) at each m⁶A site, quantifying sequence-dependent allelic differences in expression and methylation, respectively (see "Methods"). Interestingly, the allelic imbalance levels in IP samples were significantly higher than in untreated input samples (Fig. EV4A), suggesting stronger allelic biases in m⁶A methylation than in expression. Further exploration revealed a contrasting tendency of allelic bias direction in m⁶A methylation compared to the expression (Fig. 4A), which consistently manifested across all 14 samples (Fig. EV4B). We then quantified the difference in sequence-dependent allelic bias between m⁶A expression and methylation (calculated as $ARR_{c57,ip} - ARR_{c57,input}$). This analysis revealed a remarkably widespread bias for the opposing allele in m⁶A methylation compared to expression (Figs. 4B and EV4C). For instance, m⁶A sites with C57-biased expression mostly exhibited a weakened bias for the C57 allele in methylation. This reveals transcriptome-wide contrasting sequence-dependent preferences between allelic m⁶A methylation and expression, implying potential correlations between seq-ASm⁶A and sequence-dependent ASE.

In an effort to better understand the connection between seq-ASm⁶A and sequence-dependent ASE, we investigated all testable m⁶A sites exhibiting sequence-dependent ASE (see "Methods").

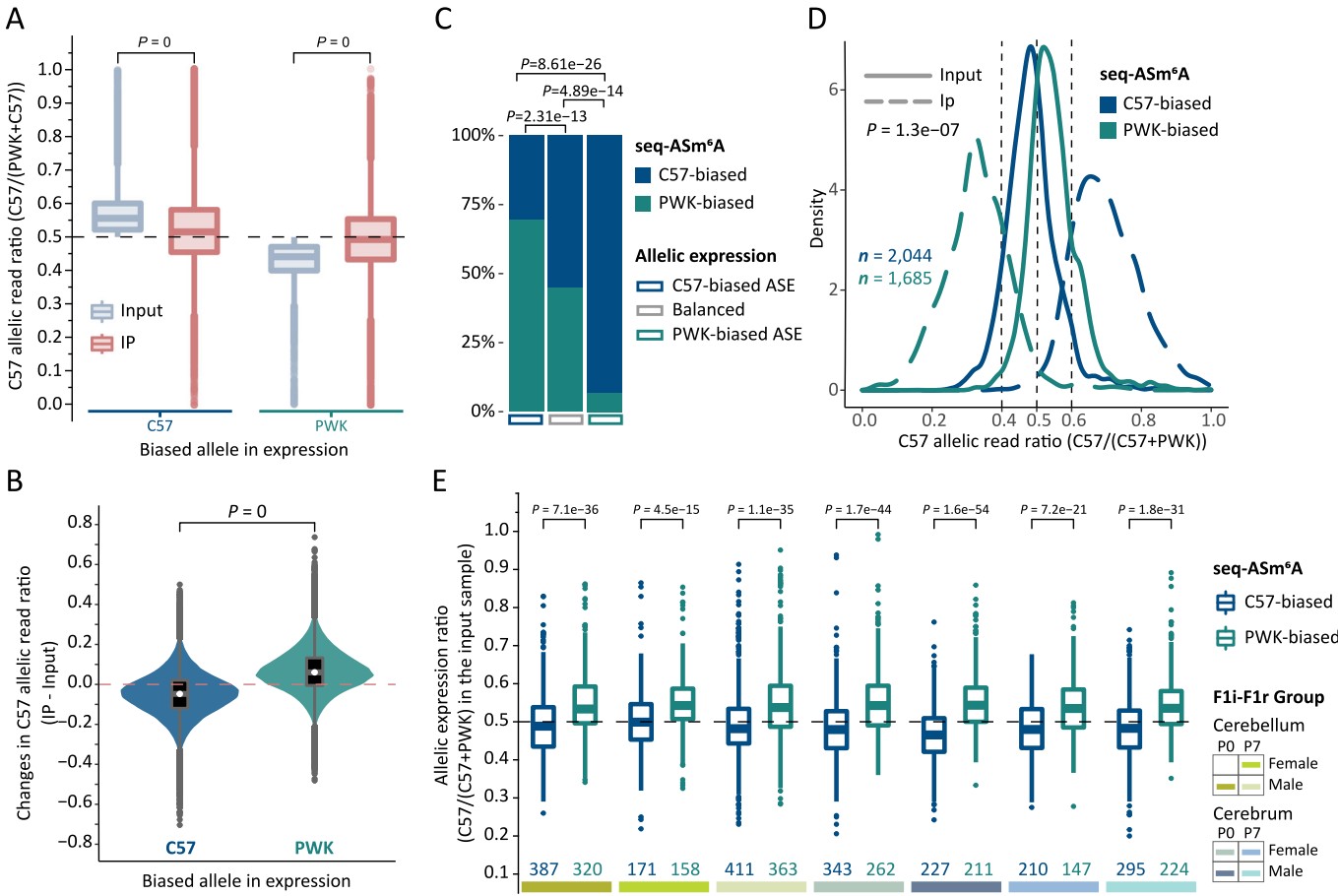

**Figure 4. Sequence-dependent ASm⁶A methylation negatively associated with ASE.**

(A) Box plots comparing C57 allelic read ratios between untreated input and IP samples. Statistical analysis utilized the two-sided paired Wilcoxon rank-sum test. (B) Distributions of differences in C57 allelic read ratios between untreated input and IP samples. White circle indicates median, and violin-shaped areas depict kernel density estimates of data distribution. Statistical analysis employed the two-sided Wilcoxon rank-sum test. (C) Stacked bar charts showing the proportions of C57- and PWK-biased seq-ASm⁶As across three categories of seq-ASm⁶A sites exhibiting distinct allelic expression patterns. Statistical analysis utilized the two-sided Pearson's chi-squared test. (D) Density plots comparing C57 allelic read ratios between untreated input and IP samples for seq-ASm⁶A sites. Statistical analysis employed the one-tailed paired Student's t test. (E) Box plots comparing allelic expression ratios in C57-biased and PWK-biased seq-ASm⁶A sites. The numbers along the x axis represent the count of seq-ASm⁶A sites. Statistical analysis utilized the two-sided Wilcoxon rank-sum test. (A, B, E) The top, middle, and bottom lines of the box represent the upper quartile (Q3), median, and lower quartile (Q1), respectively. The upper whisker extends to the maximum value provided it is not larger than $(Q3 + 1.5 \times IQR)$ (where $IQR = Q3 - Q1$), while the lower whisker extends to the minimum value provided it is not smaller than $(Q1 - 1.5 \times IQR)$. Data points beyond the whiskers are considered outliers and are plotted individually. Source data are available online for this figure.

Interestingly, we found a significant enrichment of these sites within the seq-ASm⁶A set compared to the non-seq-ASm⁶A set (Fig. EV4D). Moreover, among seq-ASm⁶A sites showing sequence-dependent ASE, 70% of these sites exhibited opposing allelic bias directions between methylation and expression (Fig. 4C). These observations suggested an intriguing hypothesis that seq-ASm⁶A might regulate ASE through allele-specific mRNA decay, a mechanism consistent with the predominant and well-established role of m⁶A in regulating mRNA stability (Lee et al, 2020, Murakami and Jaffrey, 2022, Shi et al, 2017, Zaccara and Jaffrey, 2020). To further test this hypothesis, we compared the $ARR_{c57,input}$ and $ARR_{c57,ip}$ values of all seq-ASm⁶A sites. Remarkably, we observed a significantly higher degree of allelic imbalances in methylation compared to expression, with ~70% of seq-ASm⁶As exhibiting opposing allelic bias directions between methylation and expression (Fig. 4D). Furthermore, we identified significant

differences in allelic expression between C57- and PWK-biased seq-ASm⁶A sites, with C57-biased sites showing a stronger preference for PWK allele in expression, and vice versa (Fig. 4E). These findings, in combination with previous studies, support a model in which seq-ASm⁶A regulates ASE by negatively impacting gene expression, potentially through allele-specific mRNA degradation. However, this hypothesis requires further rigorous experimental validation.

## Parent-of-origin effects on allelic m⁶A methylation

In mammals, imprinted genes exhibit preferential expression from either the maternal or paternal allele due to parent-of-origin effects. A key question arises: can these parental influences be transmitted to m⁶A modification, resulting in parent-ASm⁶A? Our stringent criteria identified no parent-ASm⁶A instances within imprinted

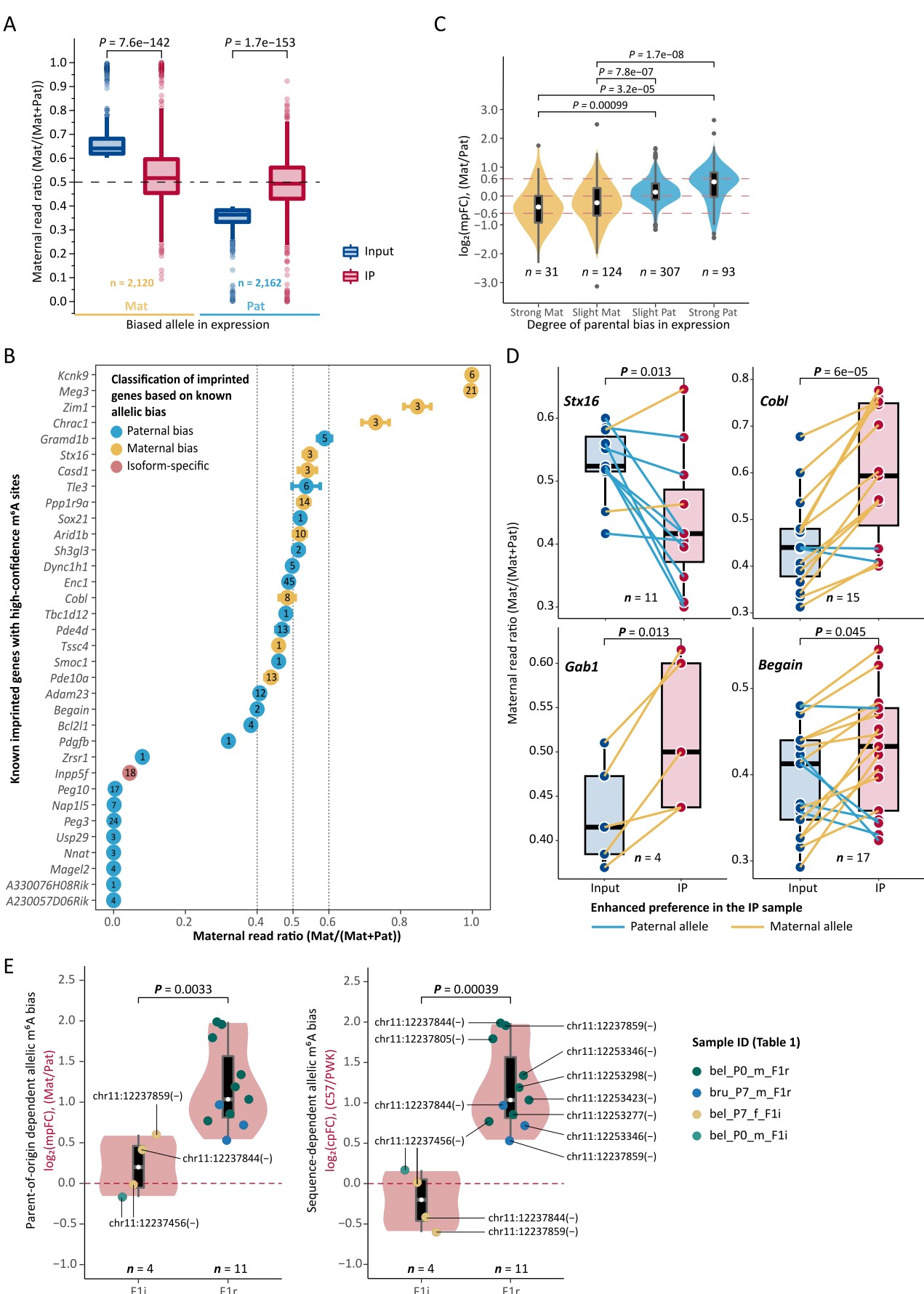

**Figure 5. Parent-of-origin effects on allelic m⁶A methylation.**

(A) Box plots comparing maternal allelic read ratios between input and IP samples for m⁶A sites showing parent-of-origin dependent ASE. Sites are divided into two groups: maternal expression bias ($n = 2120$) and paternal expression bias ($n = 2162$). Statistical analysis utilized the two-sided paired Wilcoxon rank-sum test. (B) Imprinted genes harboring high-confidence m⁶A sites in the P0 female cerebrum group. Each point represents a gene, color-coded by its reported imprinted category and labeled with its m⁶A site count. Maternal expression ratio is shown as mean ± standard error across all m⁶A sites within each gene. The genes are ranked by maternal read ratio. The dotted lines represent the cutoffs for identifying m⁶A sites showing parent-of-origin-dependent ASE (see "Methods"). (C) Distributions of parent-of-origin-dependent allelic m⁶A difference ($\log_2(mpFC)$) across allelically detectable m⁶A sites located in imprinted genes. Sites were categorized into four groups based on their maternal read ratio in untreated input samples (see "Methods"). White circle indicates median, and violin-shaped areas depict kernel density estimates of data distribution. (D) Box plots showing differences in maternal allelic read ratio between untreated input and IP samples for allelically detectable m⁶A sites in known imprinted genes. Four representative genes are shown, with the count ($n$) of m⁶A sites labeled in the plot. Statistical significance was assessed using the two-tailed paired Student's $t$ test. (E) Comparison of allelic m⁶A differences at detectable sites within the *Cobl* gene between F1i and F1r samples. Left: parent-of-origin-dependent allelic differences using $\log_2(mpFC)$ values; Right: sequence-dependent allelic differences using $\log_2(cpFC)$ values. Higher reproducibility between F1i and F1r is observed in the left plot. (A, C–E) The top, middle, and bottom lines of the box represent the upper quartile (Q3), median, and lower quartile (Q1), respectively. The upper whisker extends to the maximum value provided it is not larger than ($Q3 + 1.5 \times IQR$) (where $IQR = Q3 - Q1$), while the lower whisker extends to the minimum value provided it is not smaller than ($Q1 - 1.5 \times IQR$). Data points beyond the whiskers are considered outliers and are plotted individually. (C, E) Statistical analysis employed the two-tailed Student's $t$ test. Source data are available online for this figure.

genes, although we did detect certain seq-ASm⁶A sites in these regions. This observation prompted us to speculate that the widespread influence of *cis*-regulation may have obscured the detection of parent-ASm⁶A sites. Specifically, for an m⁶A site influenced by both sequence context and parental effects, the sequence-dependent strain-specific bias favoring opposing parental alleles in the paired F1i and F1r samples could undermine the reproducibility of parental effects between the two samples. In other words, the sequence-dependent bias might mask the potential presence of genuine parent-of-origin influences on m⁶A methylation.

To explore parental preferences in allelic methylation for m⁶A sites exhibiting parent-of-origin-dependent ASE, we introduced paired allelic read ratios, $ARR_{mat,input}$ and $ARR_{mat,ip}$, for each m⁶A site (see "Methods"). These metrics quantify allelic preferences for the maternal allele in paired untreated input (expression) and IP (methylation) samples. For each F1i-F1r group, m⁶A sites showing parent-of-origin-dependent ASE were identified based on the $ARR_{mat,input}$ values in both F1i and F1r samples (see "Methods"; Fig. EV5A). Notably, we found that higher allelic m⁶A levels correlated to lower allelic expression levels, in a parent-of-origin dependent manner (Fig. 5A). Additionally, the $ARR_{c57,input}$ and $ARR_{c57,ip}$ values (for quantifying sequence-dependent allelic preferences) of these sites showed no significant differences (Fig. EV5B), suggesting that parental influences dominate over *cis*-regulatory effects at m⁶A sites exhibiting parent-of-origin-dependent ASE. We then quantified the difference in parent-of-origin-dependent allelic bias between methylation and expression (calculated as $ARR_{mat,ip} - ARR_{mat,input}$) at these sites. A widespread bias for the opposing parental allele in m⁶A methylation compared to expression were observed (Fig. EV5C). These findings demonstrate a strong, parent-of-origin-dependent inverse correlation between allelic m⁶A methylation and expression levels.

To further investigate the impact of parental effects on m⁶A sites within imprinted genes, we focused on genes with documented parentally biased expression and harboring high-confidence m⁶A sites. Given the developmental stage and tissue-specific ASE observed in most imprinted genes, more than half of these genes in our samples exhibit moderate parental bias, with their average $ARR_{mat,input}$ values approaching 50%. Remarkably, all genes with pronounced parental expression bias displayed expression favoring the expected allele, as previously reported (Figs. 5B and EV5D). We then selected allelically

detectable m⁶A sites within imprinted genes and categorized them into four groups based on their $ARR_{mat,input}$ values: strong maternal, mild maternal, mild paternal, and strong paternal expression biases (see "Methods"). Notably, we observed a distinct inverse correlation between parental allelic expression and methylation biases at these m⁶A sites (Fig. 5C). Specifically, m⁶As with a stronger maternal expression bias exhibited a more pronounced paternal methylation bias, and vice versa (Fig. 5C). These findings underscore the crucial role of parent-of-origin effects on m⁶A methylation within certain imprinted genes, with a tendency to favor the parental allele less preferred in allelic expression.

Further examination of specific imprinted genes provided valuable insights into the divergent parental preferences between m⁶A methylation and expression. For instance, the imprinted gene syntaxin 16 (*Stx16*), known for its maternally biased expression in the mouse cerebellum (Crowley et al, 2015; Perez et al, 2015), showed allelic expression preference for the maternal allele across most allelically detectable m⁶A sites within this gene (Fig. 5D). Notably, most of these m⁶A sites showed a stronger paternal allele bias in methylation than in expression (Fig. 5D). This opposing pattern was also observed in imprinted genes with known paternally biased expression, where m⁶A sites showed an elevated maternal read ratio in the IP samples compared to the paired untreated input samples (Fig. 5D). This pattern, with opposing parental preferences between m⁶A methylation and expression, was evident in both F1i and F1r samples, suggesting that sequence-dependent effects were unlikely to be the primary driver (Fig. EV5E).

Furthermore, we observed instances in which both sequence and parent-of-origin effects jointly influenced m⁶A sites located in the same imprinted gene. For example, m⁶A sites on the cordon-bleu WH2 repeat (*Cobl*) displayed methylation preference for both C57 and maternal alleles (Fig. 5E). Despite *Cobl*'s reported tissue-specific maternally biased expression in mouse yolk sac (Shiura et al, 2009), most *Cobl* m⁶A sites showed paternally biased expression in our samples (Fig. 5E). Moreover, most of these sites exhibited maternal preference in allelic methylation, contrasting with paternal preference in expression. However, these sites were also influenced by sequence-dependent effects, displaying allelic bias for the C57 allele (Fig. 5E). By analyzing the sites allelically detectable in both F1i and F1r samples (chr11:12237456(-), chr11:12237844(-) and chr11:12237859(-)), we observed higher reproducibility in the $\log_2(mpFC)$ values between F1i and F1r

samples compared to the $\log_2(cpFC)$ values (Fig. 5E). This indicates that the maternal methylation preference of these m⁶A sites appeared more dominant than the preference for C57 allele. Collectively, these findings underscore parent-of-origin effects on allelic m⁶A methylation, potentially further modulating parent-of-origin-dependent ASE of m⁶A-marked genes.

## Discussion

In mammals, sequence- and parent-of-origin-dependent ASE arise from genetic or epigenetic variations between alleles, often mediated by allele-specific events such as ASM (Cleary and Seoighe, 2021, Onuchic et al, 2018, Xie et al, 2012). Our study demonstrates that ASm⁶A modifications introduce an additional layer of complexity to ASE regulation, providing insights into the mechanisms governing their establishment and effects on post-transcriptional regulatory processes (Fig. 6). Through comprehensive analysis of ASm⁶A in highly divergent mouse crosses, we reveal widespread sequence-dependent allelic m⁶A imbalances with remarkable reproducibility across diverse samples. Exploration of relevant cis-acting variants highlights the crucial role of cis-regulatory mechanisms in shaping allele-specific m⁶A modification patterns, thereby exerting allele-specific regulatory influence over the fate of m⁶A-marked transcript. Notably, our findings reveal a strong negative correlation between allelic m⁶A methylation and expression, suggesting a potential role of ASm⁶A in regulating ASE through negative effects on gene expression.

Among the identified ASm⁶A sites, the majority were classified as seq-ASm⁶A, highlighting the predominant influence of sequence context on allelic m⁶A methylation. The prevalence of sequence-dependent allelic m⁶A imbalance aligns with findings from previous

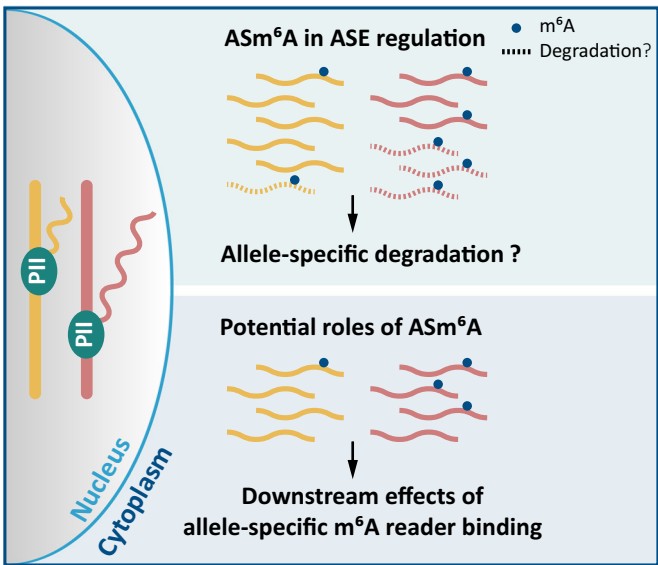

**Figure 6. Potential roles of ASm⁶A.**

Our analysis establishes a connection between allelic m⁶A methylation and expression, supporting a potential role for ASm⁶A in regulating ASE through allele-specific mRNA degradation. Considering the established functional mechanisms of m⁶A, ASm⁶A may influence other downstream allele-specific events through allele-specific binding of m⁶A readers.

ASE (Crowley et al, 2015) and ASM studies (Cleary and Seoighe, 2021; Xie et al, 2012), emphasizing the widespread cis-regulatory effects of genetic variants on various allele-specific events. For seq-ASm⁶A sites, we observed substantial directional agreements (>91%) in allelic m⁶A preferences and strong positive correlations in allelic imbalance levels among samples. This consistency is also supported by previous studies on m⁶A QTLs in humans, which demonstrated that the directions of QTL effects rarely change across diverse tissues (Xiong et al, 2021; Zhang et al, 2020b). Taken together, cis-acting variants and their associated regulators exhibit remarkable stability in determining the direction of their effects on m⁶A levels, although the intensity of these effects may vary in diverse cellular environments.

We identified potential regulatory loci influencing m⁶A levels by conducting an enrichment analysis of SNP positions flanking seq-ASm⁶A sites. As expected, m⁶A sites with variants at motif loci showed significant enrichment in seq-ASm⁶A sites. However, >90% of the seq-ASm⁶A sites have no variant in the motif region, suggesting widespread cis-regulatory effects from other loci. All the identified cis-acting loci were within the flanking 50-nt regions of seq-ASm⁶A sites, underscoring the crucial role of adjacent sequence context in m⁶A regulation (Shachar et al, 2024). Future studies are needed to identify the regulators associated with these loci and responsible for regulating m⁶A levels. The cis-acting loci discovered in our study represent a crucial resource for future investigations into m⁶A regulation.

In mammals, parent-of-origin effects play a crucial role in regulating the allele-specific expression of imprinted genes, primarily through allele-specific DNA methylation and histone modifications (Barlow, 2011; Delaval and Feil, 2004; Inoue et al, 2017; Tucci et al, 2019; Zink et al, 2018). Our study extends this understanding by revealing parent-of-origin effects on m⁶A modifications within imprinted genes. These effects result in divergent parental preferences between allelic m⁶A methylation and expression, representing the first discovery of parental influence on post-transcriptional modifications. Deciphering the interplay between parent-of-origin effects and m⁶A modifications holds significant implications for the fields of epigenetics and epitranscriptomics. Additionally, since imprinted gene expression is predominantly regulated in the transcriptional layer (Barlow, 2011; Delaval and Feil, 2004; Tucci et al, 2019), future studies examining allelic m⁶A modifications in nascent or nuclear mRNA may offer a deeper understanding of parental effects on ASm⁶A methylation and imprinted gene regulation.

Exploring the potential downstream effects of ASm⁶A will enhance our understanding of m⁶A functions. Previous studies show that the major function of mammalian m⁶A is to promote mRNA decay (Lee et al, 2020; Murakami and Jaffrey, 2022; Shi et al, 2017; Zaccara and Jaffrey, 2020). Our analysis of allelic imbalance in m⁶A methylation and expression revealed a potential role for ASm⁶A in regulating ASE by negatively affecting gene expression, consistent with the well-established function of m⁶A. Based on our findings and the known roles of m⁶A in mRNA regulation (Frye et al, 2018; He and He, 2021; Lee et al, 2020; Murakami and Jaffrey, 2022; Roundtree et al, 2017; Shi et al, 2019; Zaccara and Jaffrey, 2020; Zaccara et al, 2019), we propose two potential mechanisms of ASm⁶A function: (1) allele-specific mRNA degradation; and (2) allele-specific reader binding, potentially leading to downstream allele-specific events (Fig. 6). These hypotheses require further rigorous experimental testing. For example, the first could be tested by measuring transcript abundance with and without allele-specific

m⁶A inhibition. The second proposes that allelic differences in m⁶A methylation could alter allelic binding of m⁶A reader proteins, potentially resulting in additional downstream allele-specific events (including, but not limited to, splicing and translation). Future studies are needed to identify downstream allele-specific events and their associations with ASm⁶A.

m⁶A plays a critical role in mouse cerebellum and cerebrum development. Dynamic regulation of METTL3-mediated m⁶A methylation is associated with cerebellar development (Ma et al, 2018; Wang et al, 2018). Additionally, m⁶A writers and readers show a decreasing trend with age during cerebellum development, with the highest level in P7, suggesting a critical role for m⁶A in early cerebellar development (Jiang et al, 2022). In the cerebral cortex, m⁶A writers and erasers are ubiquitously expressed across different layers, revealing m⁶A functions in diverse neural cell types (Chang et al, 2017). In our study, we identified numerous m⁶A sites influenced by genetic variation during early development (P0 and P7) in both cerebellum and cerebrum. These sites and the related *cis*-elements may participate in regulating relevant developmental processes and potentially impact phenotypic outcomes. However, existing studies are predominantly based on data from the C57 strain, lacking a systematic investigation of m⁶A modifications in the PWK strain. We expect that future studies will obtain m⁶A modification data from both parental strains and integrate this with phenotypic information to further explore the biological roles of ASm⁶A in mammals. Interestingly, despite the known abundance and functional importance of m⁶A in *Arabidopsis*, allele-specific analysis in *Arabidopsis* hybrids revealed little allelic imbalance in m⁶A methylation (Xu et al, 2021). This contrasts with the findings in mammals, suggesting that the regulation of m⁶A may differ significantly between plants and mammals, representing an important area for future study.

Overall, our study represents the pioneering exploration of transcriptome-wide allele-specific m⁶A profiles, revealing both sequence- and parent-of-origin-dependent effects on allelic m⁶A. The resulting allele-specific m⁶A profiles, ASm⁶A sites, *cis*-acting loci, and parental effects on m⁶A have profound implications for understanding genetic and epigenetic mechanisms controlling m⁶A modification. The proposed roles of ASm⁶A in regulating ASE offer valuable insights for future studies aimed at exploring how genetic and epigenetic effects on m⁶A further impact post-transcriptional gene regulatory processes.

# Methods

### Reagents and tools table

| Reagent/resource | Reference or source | Identifier or catalog number |
|---|---|---|
| **Experimental models** | | |
| C57BL/6J (*M. musculus*) | Vital River | – |
| PWK/PhJ (*M. musculus*) | Gift from Prof. Qi Zhou | – |
| **Recombinant DNA** | | |
| **Antibodies** | | |
| N6-Methyladenosine (m6A) (D9D9W) Rabbit mAb | CST | 56593 |

| Reagent/resource | Reference or source | Identifier or catalog number |
|---|---|---|
| **Oligonucleotides and other sequence-based reagents** | | |
| rRNA depletion probe (sequences refer to Adiconis et al, 2013) | Sangon biotech | – |
| 5′-adapter (Table EV4) | Hippo biotech | – |
| 3′-adapter (Table EV4) | Sangon biotech | – |
| 3′-adapter encoded with 6-nt barcode (Table EV4) | Sangon biotech | – |
| RT primer (Table EV4) | Sangon biotech | – |
| **Chemicals, enzymes and other reagents** | | |
| Trizol reagent | Invitrogen | 15596018CN |
| Dynabeads mRNA Purification Kit | Invitrogen | 61006 |
| Dynabeads Protein G | Invitrogen | 10004D |
| RNase H | NEB | M0297 |
| DNase I (RNase-free) | NEB | M0303 |
| Oligo clean & concentrator – 5 | Zymo Research | D4060 |
| RNA clean & concentrator – 5 | Zymo Research | R1014 |
| 5′DNA Adenylation Kit | NEB | E2610 |
| RNA Fragmentation Reagents | Invitrogen | AM8740 |
| T4 Polynucleotide Kinase | NEB | M0201 |
| Recombinant ribonuclease inhibitor | Takara Bio | 2313 |
| T4 RNA ligase 2, truncated KQ | NEB | M0373 |
| Lambda Exonuclease | NEB | M0262 |
| 5′ Deadenylase | NEB | M0331 |
| Buffer RLT | QIAGEN | 9216 |
| T4 RNA Ligase 1 (ssRNA Ligase) | NEB | M0204 |
| HiScript III 1st Strand cDNA Synthesis Kit ( + gDNA wiper) | Vazyme | R312 |
| 2×KAPA HiFi Hot Start Ready Mix | KAPA Biosystems | KK2602 7958935001 |
| HiScript II One Step qRT-PCR SYBR Green Kit | Vazyme | Q221-01 |
| VAHTS RNA clean beads | Vazyme | N412 |
| VAHTS DNA clean beads | Vazyme | N411 |
| Sodium chloride 5 M | Sigma-Aldrich | 7647-14-5 |
| Tris-HCl (pH 7.4) | Sigma-Aldrich | T2663 |
| Igepal CA-630 | Sigma-Aldrich | I8896 |
| Glyoxal solution | Sigma-Aldrich | 50649 |
| Sodium nitrite | Sigma-Aldrich | 7632-00-0 |
| **Software** | | |
| GLORI-tools v.1.0 | https://github.com/liucongcas/GLORI-tools Liu et al, 2023 | |

| Reagent/resource | Reference or source | Identifier or catalog number |
|---|---|---|
| SNPsplit v.0.3.2 | Krueger and Andrews, 2016 | |
| fastq-multx v.1.4.3 | https://expressionanalysis.github.io/ea-utils/ | |
| cutadapt v.2.10 | Martin, 2011 | |
| fastp v.0.21.0 | Chen et al, 2018 | |
| hisat2 v.2.1.0 | Kim et al, 2019 | |
| SAMtools v.1.15.1 | Li et al, 2009 | |
| gencore v.0.16.0 | Chen et al, 2019 | |
| MACS2 v.2.1.2 | Zhang et al, 2008 | |
| bedtools v.2.26.0 | Quinlan, 2014; Quinlan and Hall, 2010 | |
| featureCounts v.1.6.0 | Liao et al, 2014 | |
| metaPlotR | Olarerin-George and Jaffrey, 2017 | |
| Other | | |

## Ethical statement

All mouse-related experiments were conducted at the Institute of Genetics and Developmental Biology, Chinese Academy of Sciences, and reviewed by the institutional Animal Care and Use Committee.

## Mice

C57BL/6J (C57) and PWK/PhJ (PWK) mice were bred inside a specific-pathogen-free barrier with a 12-hour light-dark cycle, $22 \pm 1\,°C$ room temperature, ~50% humidity, and free access to water and food. To generate reciprocal F1 hybrids of the two strains, 8-week-old male or female C57 mice were co-housed with the opposite sex of 8-week-old PWK mice. Newborn pups were collected on postnatal day 0 (P0) and day 7 (P7).

## Tissue collection

P0 and P7 mice were sacrificed by cervical dislocation and were immediately decapitated. Mouse brains were dissected under a stereomicroscope and were washed in cold 1× Hank's Balanced Salt Solution (HBSS) for three times to remove blood and body fluid. Then, cerebellum and cerebrum tissues were collected and separately homogenized in the presence of TRNzol Universal reagent (TianGen, #DP424).

## RNA preparation

Total RNA was extracted in Trizol reagent following the manufacturer's protocol. Poly(A)$^+$ RNA was then isolated using Dynabeads mRNA Purification Kit (Invitrogen, #61006). For ribo-depleted total RNA, we employed a previously published method (Adiconis et al, 2013). Briefly, rRNA was hybridized with a set of rRNA probes and subsequently digested by RNase H (NEB, #M0297S). After digestion, the rRNA probes were removed using DNase I (NEB, #M0303S) and finally RNA was purified using 2.2×

VAHTS RNA clean beads (Vazyme, #N412). The concentration of RNA samples was determined via a Qubit Fluorometer.

## Multiplexed m⁶A-seq

### Paired F1i-F1r samples

Allele-specific m⁶A profiling was conducted using multiplexed m⁶A-seq in two tissues across two early developmental stages, for enhanced comparability between samples (Table EV1): the P0 cerebellum ($n = 2$), P7 cerebellum ($n = 4$), P0 cerebrum ($n = 4$), and P7 cerebrum ($n = 4$). The choice of these two brain tissue types was driven by the profound implications of ASE in brain function and disease, particularly in the context of parent-of-origin dependent ASE (Gregg et al, 2010; Huang et al, 2018; Kravitz and Gregg, 2019; Perez et al, 2015). To distinguish between sequence- and parent-of- origin-dependent allele-specific m⁶A methylation, m⁶A profiling was performed on paired samples: F1i and F1r. F1i mice were produced by crossing a PWK mother with a C57 father, while F1r mice resulted from the reciprocal cross involving a C57 mother and a PWK father. Each F1i-F1r pair was matched for age and sex to minimize potential confounding variables (Table EV1).

### Library preparation

RNA purification was conducted using RNA clean & concentrator kits-5 (Zymo Research, #R1016), unless otherwise specified. The 3′-adapters with 6-nt barcode and 5′-adapters with 8-nt unique molecular identifier (UMI) were synthesized by Sangon Biotech and Hippo Bio, respectively (Table EV2). Synthesized 3′-adapter encoded with 6-nt barcode was pre-adenylated using 5′ DNA Adenylation Kit (NEB, E2610S). For each sample from RNA preparation step, 200 ng mRNA/ribo-depleted RNA of each sample was fragmented to ~150–200 nt using RNA fragmentation reagents (Invitrogen, #AM8740). Fragmented RNA was purified and then repaired using T4 Polynucleotide Kinase (NEB, #M0201S). After cleanup, end-prepared RNA was ligated to adenylated 3′-adapter by T4 RNA Ligase 2, truncated KQ (NEB, #M0373S) incubated at 25 °C for 2 h, followed by 4 °C overnight. To remove redundant adapter, 2 μl of Lambda Exonuclease (NEB, #M0262S) and 1 μl of 5′ Deadenylase (NEB, #M0331S) were added to each ligation mix. The reaction was incubated at 30 °C for 30 min, then at 37 °C for 30 min and finally 70 °C for 10 min to inactivate the enzymes. Subsequently, the samples were pooled to perform cleanup together and 10% of the sample pool was reserved as untreated input sample. Multiplexed m⁶A immunoprecipitation (IP) was conducted two rounds, based on a published work (Dierks et al, 2021). Specifically, for each IP round, 25 μL Dynabeads Protein G (ThermoFisher, #10004D) were coated with 2.5 μL m⁶A antibody (CST, #56593). In the first round, the pooled sample was heated at 70 °C for 2 min, cooled on ice for 2 min, and incubated with antibody-coated beads for 2 h at 4 °C with rotation. Beads were washed twice with 1× IPP buffer, low-salt buffer (50 mM NaCl, 0.01% Igepal CA-630, 10 mM Tris-HCl, pH 7.5) and high-salt buffer (500 mM NaCl, 0.01% Igepal CA-630, 10 mM Tris-HCl, pH 7.5), respectively. The IP product was eluted using Buffer RLT (QIAGEN, #79216) following purification. The second round of IP was conducted identically to the first. After the two rounds of IP, untreated input and IP samples were respectively ligated to pre-heated 5′-adapter with T4 RNA Ligase 1 (NEB, #M0204S), and subjected to reverse transcription using HiScript III 1st Strand cDNA Synthesis Kit (Vazyme, #R312-01) and RT primer without cleanup. Library

amplification was conducted with KAPA HiFi HotStart ReadyMix (KAPA Biosystems, #KK2601) with the universal primer and the indexed primer (Vazyme, #N814). Finally, the amplified libraries were purified with 0.9× VAHTS DNA Clean Beads (Vazyme, #N411) and were sent for 2 × 150-base-pair paired-end sequencing.

## GLORI

### Library preparation
We conducted glyoxal and nitrite-mediated deamination treatment on two RNA samples, one from the cerebellum and another from the cerebrum of a female P0 F1r mouse (Table EV1), following the published method (Liu et al, 2023) with slight modifications. Specifically, we increased incubation temperature during the deamination step to 25 °C instead of 16 °C. For library construction, 50 ng of the GLORI-treated RNA sample was firstly repaired and ligated to pre-adenylated 3′-adapter as described in the multiplexed m6A-seq section (Table EV2). After the redundant adapter was digested by Lambda Exonuclease (NEB, #M0262S) and 5′ Deadenylase (NEB, #M0331S), the RNA samples were ligated to a pre-heated 5′-adapter without cleanup. Reverse transcription was then carried out using HiScript III 1st Strand cDNA Synthesis Kit (Vazyme, #R312-01) and RT primer. Finally, the library was amplified using Q5 High-Fidelity DNA Polymerases with GC enhancer (NEB, #M0491S) and subjected to size selection using a 6% polyacrylamide gel, retaining fragments longer than 150 nt. The gel-extracted fragments were re-amplified using the same condition to reach sequencing concentration. The libraries were sent for 2 × 150-base-pair paired-end sequencing on the Illumina NGS platform with PhiX included to balance base bias.

### Single-base m6A identification
Calling m6A sites was conducted using the pipeline and code developed by Liu et al (GLORI-tools v.1.0; https://github.com/liucongcas/GLORI-tools) (Liu et al, 2023), with GENCODE (v26; downloaded from https://www.gencodegenes.org/mouse/) as annotation. For each sample, two sets of m6A sites were obtained by using two reference genomes for read mapping: the mm10 (C57 strain) and PWK-mm10 genomes. The PWK-mm10 genome was generated by replacing each SNP site in the mm10 genome with the corresponding base from the PWK genome. Calling m6A sites using this genome could detect PWK-specific adenosines with m6A modification. For each sample, m6A sites with a methylation rate >0.1 and read count >8 in the union of these two sets were retained for downstream analysis.

## m6A-seq data analysis

### Preparation of SNPs and the N-masked genome
SNPs between C57 and PWK mouse strains were extracted from the Mouse Genomes Project database (ftp://ftp-mouse.sanger.ac.uk). The N-masked mouse genome, where all SNPs between the C57 and PWK genomes were replaced by "N", was prepared based on the mm10 reference genome using SNPsplit_genome_preparation function of SNPsplit package (v. 0.3.2) (Krueger and Andrews, 2016). SNPs retained by SNPsplit, which passed high-confidence filters (Krueger and Andrews, 2016), were used for masking and downstream SNP-associated analysis.

### Read alignment
Paired-end reads of multiplexed m6A-seq libraries were demultiplexed into individual samples using fastq-multx (v.1.4.3; https://expressionanalysis.github.io/ea-utils/) with one mismatch allowed. Adapters at the 3′ end of the raw reads were trimmed with cutadapt (v.2.10) (Martin, 2011). The reads were then processed, and UMIs were extracted using fastp (v.0.21.0) (Chen et al, 2018) with the following parameters: -l 15 -U --umi_loc=read1 --umi_len=10 --umi_prefix UMI. The remaining reads were aligned to the N-masked genome to minimize reference bias. The alignment was performed using hisat2 (v.2.1.0) (Kim et al, 2019), with GENCODE (v26) as annotation, and with the following parameters: -k 1 --rna-strandness FR --no-unal --no-softclip. Properly paired and mapped reads (-f 3) were retained with SAMtools (v.1.15.1) (Li et al, 2009). The PCR duplicates were removed by gencore (v.0.16.0) (Chen et al, 2019).

### m6A peak detection
For stranded peak calling, reads in each sample were then divided into two groups based on the XS tag (XS:A:- or XS:A:+) in the SAM file. Peak calling was performed independently on both groups using MACS2 (v.2.1.2) (Zhang et al, 2008) with the parameters of --nomodel and --extsize 80. Peak processing was performed using bedtools (v.2.26.0) (Quinlan, 2014; Quinlan and Hall, 2010).

### Allele-specific read assignment
The alignments were split into respective alleles using SNPsplit (v.0.3.2) (Krueger and Andrews, 2016) with options: --paired --no_sort. Allelically aligned reads for each m6A site were counted by featureCounts (v.1.6.0, -p -B -C -s 1) (Liao et al, 2014).

## High-confidence m6A identification and quality control

For each m6A-seq sample, we identified high-confidence single-base m6A sites. This was done by overlapping m6A peaks in the specific sample with GLORI-detected m6As in the corresponding tissue. We evaluated the quality of the identified m6As using two well-recognized metrics: the distribution along mRNA and the consensus motif. Metagene distributions were analyzed using metaPlotR (Olarerin-George and Jaffrey, 2017), and motif enrichment analysis was performed using the R package ggseqlogo (v.0.1).

## Strain-specific adenine sites with m6A modification

Single-base m6As located at SNP positions indicate strain-specific adenine sites with m6A modification. These sites were identified by intersecting all detected m6A sites with high-confidence SNPs, which play a crucial role in downstream analyses. When performing allele-specific quantification on other m6A sites, reads covering these SNP-overlapping m6As were excluded to avoid potential allelic bias in the quantification of neighboring m6As within the same fragment in the IP sample. Additionally, these loci served as internal positive controls for ASm6A, facilitating the analysis of the distribution and fluctuations of quantification metrics for allelic m6A imbalance (refer to the "ASm6A identification and classification" section for details).

## Allele-specific m⁶A quantification

A set of m⁶A-seq data comprises a pair of untreated input (regular RNA sequencing) and IP samples. Raw counts of allelically aligned reads in both samples were used for allele-specific m⁶A quantification. As previously stated, reads covering the SNP-overlapping m⁶As were excluded to mitigate potential allelic bias. For a given m⁶A site $i$, we derived four allele-specific m⁶A levels denoted as $L_{ij}$ and quantified as:

$$L_{ij} = \log\left(\frac{p_{ij}/t_{ij}}{P_j/T_j}\right)$$

where $j \in \{'c57', 'pwk', 'mat', 'pat'\}$ represents the strain or parent-of-origin of the allele. $t_{ij}$ and $p_{ij}$ denote the allele-$j$-specific read counts of m⁶A site $i$ in the untreated input and IP samples, respectively. Four pairs of $t_{ij}$ and $p_{ij}$ values were derived for each m⁶A $i$. Correspondingly, for each untreated input or IP sample, we defined four allele-specific library sizes based on the total counts of allele-$j$-specific reads, denoted by $T_j$ or $P_j$. For the untreated input sample, the four library sizes were denoted as $T_{c57}$, $T_{pwk}$, $T_{mat}$, and $T_{pat}$, while for the IP sample, they were $P_{c57}$, $P_{pwk}$, $P_{mat}$, and $P_{pat}$. Importantly, each allele could be assigned two $j$ values, representing the strain and parent-of-origin attributes. Specifically, in F1i samples: $L_{i,c57} = L_{i,pat}$, $L_{i,pwk} = L_{i,mat}$; in F1r samples: $L_{i,c57} = L_{i,mat}$, $L_{i,pwk} = L_{i,pat}$.

## Preliminary filters of m⁶A sites for allelic analysis

To ensure the high reliability of our analysis, we limited allelic m⁶A analysis exclusively to sites exhibiting adequate allelic read coverage. Based on the raw counts of allelically aligned reads, we defined three groups of m⁶A sites for downstream allele-specific analysis (Fig. EV2A):

### Allelically detectable m⁶A sites

For each m⁶A-seq dataset, allelically detectable sites were defined as those with both alleles expressed, showing detectable allelic expression and methylation levels. To identify these sites, we applied the following filtering criteria to the identified high-confidence m⁶A sites: $t_{i,c57} \geq 5$, $t_{i,pwk} \geq 5$, $t_{i,c57} + t_{i,pwk} \geq 20$, and $p_{i,c57} + p_{i,pwk} \geq 20$. Here, $t_{i,c57}$ and $p_{i,C57}$ denote the allele-C57-specific read counts of m⁶A site $i$ in the untreated input and IP samples, respectively; $t_{i,pwk}$ and $p_{i,pwk}$ denote the allele-PWK-specific read counts of m⁶A site $i$ in the untreated input and IP samples, respectively. Only m⁶A sites meeting these criteria were used in the downstream allelic analysis.

### Testable m⁶A sites in one sample

To improve the reliability and validity of ASm⁶A identification based on binomial testing (refer to the "ASm⁶A identification and classification" section for details), testable sites were selected with a higher allelic read count cutoff in the untreated input sample. These sites were derived from allelically detectable sites with $t_{i,c57} + t_{i,pwk} \geq 30$.

### Testable m⁶A sites in both samples of one F1i-F1r group

These sites were obtained by intersecting testable m⁶A sites in both samples within the same F1i-F1r group.

## Clustering and PCA of allele-specific m⁶A profiles

### Allele-specific m⁶A profiles

Based on allele-specific m⁶A quantification, we obtained two allele-specific m⁶A levels for each allelically detectable site from a single m⁶A-seq dataset. Treating data from each allele independently, we divided each original m⁶A-seq sample into two allele-specific sub-samples. This approach resulted in the derivation of two allele-specific m⁶A profiles from each initial m⁶A-seq sample. In total, 28 allele-specific m⁶A profiles were derived from the original 14 m⁶A-seq datasets (Table EV1).

### Hierarchical clustering

To explore the relationships among allele-specific m⁶A profiles, Pearson correlation coefficients were computed for each pair of sub-samples. This calculation focused on the intersection of allelically detectable m⁶A sites across all sub-samples, excluding strain-specific adenine sites. Subsequently, hierarchical clustering was performed using the Pearson correlation coefficients to group sub-samples with similar m⁶A modification patterns. Notably, attributes such as tissue, age, sex, cross, strain, and parent-of-origin were annotated, providing insights into shared regulatory factors among the analyzed allele-specific m⁶A profiles. Heatmaps were generated by the R package pheatmap v.1.0.12 (https://CRAN.R-project.org/package=pheatmap).

### PCA

PCA was employed to investigate the inherent variability and relationships among sub-samples within each tissue based on allele-specific m⁶A profiles. The analysis utilized the prcomp function in the default packages of R (v.4.0.2). Allelic m⁶A levels for the intersection of allelically detectable sites across all sub-samples were considered for each tissue.

## Quantification metrics for allelic m⁶A imbalance

We assessed allelic m⁶A imbalance by calculating the fold change (FC) between allele-specific methylation levels at each m⁶A site. Different FC forms were defined for various analytical perspectives. To quantify sequence-dependent allelic m⁶A imbalance level between alleles originating from different strains, the following quantification metric was computed for a given m⁶A site:

$$\log_2(cpFC) = \log_2\left(\frac{L_{c57}}{L_{pwk}}\right)$$

$L_{c57}$ and $L_{pwk}$ denote the allele-specific m⁶A levels for C57 and PWK alleles, respectively.

Similarly, when quantifying parent-of-origin-dependent allelic m⁶A imbalance level between alleles from different parents, the corresponding quantification metric was expressed as:

$$\log_2(mpFC) = \log_2\left(\frac{L_{mat}}{L_{pat}}\right)$$

$L_{mat}$ and $L_{pat}$ represent the allele-specific m⁶A methylation levels for maternal and paternal alleles, respectively.

To assess allelic methylation imbalance at both group and tissue levels, two quantification metrics were computed for each m⁶A site. The group-level $\log_2(cpFC)$ or $\log_2(mpFC)$ was obtained by averaging the

$\log_2(cpFC)$ or $\log_2(mpFC)$ values from F1i and F1r samples within an F1i-F1r group. The tissue-level $\log_2(cpFC)$ or $\log_2(mpFC)$ for F1i (or F1r) samples was determined as the mean across F1i (or F1r) samples within the same tissue, considering m⁶A sites with detectable allelic levels.

## Inter-sample same-strain and intra-sample different-strain allelic m⁶A imbalance

To perform a transcriptome-wide assessment of sequence-dependent effects on m⁶A, we derived two virtual negative control datasets from the original datasets for each F1i-F1r group. The negative control represents the allelic m⁶A comparison between alleles from the same strain, used to assess background allelic m⁶A imbalance level independent of strain-dependent effects. Specifically, such dataset comprises sequencing data from two alleles of the same strain (C57 or PWK). For a given m⁶A site, the allelic methylation imbalance level in the virtual dataset was also calculated based on FC, as follows:

$$I_s = \left| \log_2\left( \frac{L_{s,F1i}}{L_{s,F1r}} \right) \right|$$

where $s \in \{'c57', 'pwk'\}$ denotes the strain attribute of the allele. $L_{s,F1i}$ and $L_{s,F1r}$ represent the allele-$s$-specific levels of a given m⁶A site in the F1i and F1r samples, respectively. Here, we focused on the level of allelic methylation bias, not its direction, and therefore used absolute values. For each m⁶A site, we obtained two allelic imbalance levels, $I_{c57}$ and $I_{pwk}$, in an F1i-F1r group. We used the mean of them to quantify the inter-sample same-strain allelic m⁶A imbalance level within an F1i-F1r group:

$$Inter = \frac{I_{c57} + I_{pwk}}{2}$$

Correspondingly, for a given m⁶A site, the intra-sample different-strain allelic m⁶A imbalance level within an F1i-F1r group was quantified as follows:

$$Intra = \frac{|I_{F1i}| + |I_{F1r}|}{2}$$

Here, $I_{F1i}$ and $I_{F1r}$ represent the $\log_2(cpFC)$ values for the m⁶A site in F1i and F1r samples, respectively.

## Allelic read ratio in untreated input and IP samples

A set of m⁶A-seq data comprises paired untreated input (regular RNA sequencing) and IP samples. For a given m⁶A site, we employed allelic read ratio (ARR) to quantify its allelic coverage imbalance level in both sample types. $ARR_{c57}$ and $ARR_{mat}$ were defined to measure sequence-dependent and parent-of-origin-dependent allelic read bias levels, respectively:

$$ARR_{c57,i} = \frac{R_{c57,i}}{R_{c57,i} + R_{pwk,i}}$$

$$ARR_{mat,i} = \frac{R_{mat,i}}{R_{mat,i} + R_{pat,i}}$$

where $i \in \{'input', 'ip'\}$ denotes sample type. $R_{c57,i}$, $R_{pwk,i}$, $R_{mat,i}$, and $R_{pat,i}$ represent the read counts for C57, PWK, maternal, and paternal alleles, respectively.

In our downstream analysis, $ARR_{c57,input}$ and $ARR_{mat,input}$ were utilized to identify m⁶A sites exhibiting ASE. Based on $ARR_{mat,input}$ values, we categorized allelically detectable m⁶A sites within imprinted genes into four groups: (0.6, 1] for strong maternal bias; (0.5, 0.6] for mild maternal bias; [0.4, 0.5) for mild paternal bias; [0, 0.4) for strong paternal bias. Additionally, by comparing ARR values between paired untreated input and IP samples, we investigated the associations between allelic methylation and allelic expression. This comparison was also employed in the significance test to assess allelic differences in m⁶A methylation (refer to the "ASm⁶A identification and classification" section for details).

## ASm⁶A identification and classification

To identify ASm⁶As with significant allelic differences, we assessed three metrics for a given testable m⁶A site in each F1i-F1r group: statistical significance, allelic imbalance level, and reproducibility between paired F1i and F1r samples (Fig. 2D).

### Statistical testing
A binomial model was employed to assess the statistical significance of allelic methylation imbalance for each m⁶A site based on m⁶A-seq datasets. The read count for C57 or PWK allele in the IP sample was treated as a binomial random variable:

$$R_{c57,ip} \sim Binomial\left( n = R_{c57,ip} + R_{pwk,ip}, p = ARR_{c57,input} \right)$$

$$R_{pwk,ip} \sim Binomial\left( n = R_{c57,ip} + R_{pwk,ip}, p = ARR_{pwk,input} \right)$$

Here, $R_{c57,ip}$ and $R_{pwk,ip}$ represent the read counts in the IP sample for C57 and PWK alleles, respectively. $ARR_{c57,input}$ and $ARR_{pwk,input}$ denote the ARR values obtained from the untreated input sample for C57 and PWK alleles, respectively. The null hypothesis for the binomial test states that there is no significant allelic imbalance in methylation for a given m⁶A site, where a randomly sampled sequence read in the IP sample has a probability of $ARR_{c57,input}$ being generated from the C57 allele, and a probability of $ARR_{pwk,input}$ being generated from the PWK allele. The Benjamini–Hochberg procedure was used to control the FDR at 10%.

### Allelic imbalance level
The absolute value of $\log_2(cpFC)$ was employed to evaluate the allelic imbalance level for each testable m⁶A site. To assess stochastic fluctuations in the $\log_2(cpFC)$ values and establish an optimal threshold for identifying ASm⁶A, an analysis was conducted on the testable m⁶As located at strain-specific adenine sites, treated as positive controls for ASm⁶A. The fluctuation levels of $\log_2(cpFC)$ values were assessed using Euclidean distance as follows:

$$d_i = N_{i+} - N_{i-}$$

$$d_r = N_{r+} - N_{r-}$$

$$D = \sqrt{d_i{}^2 + d_r{}^2}$$

Here, $N_{i+}$ and $N_{r+}$ denote the medians of positive $\log_2(cpFC)$ values in F1i and in F1r samples, respectively. Similarly, $N_{i-}$ and $N_{r-}$ denote the medians of negative $\log_2(cpFC)$ values in F1i and in F1r samples, respectively. $d_i$ and $d_r$ represent the fluctuation levels of $\log_2(cpFC)$ values in the F1i and F1r samples, respectively. $D$ represents the overall fluctuation level of $\log_2(cpFC)$ values in the entire F1i-F1r group. To establish an optimal threshold for identifying ASm⁶A candidates in one sample, the average fluctuation level of $\log_2(cpFC)$ in one sample (F1i or F1r), denoted as $C$, was calculated:

$$C = \sqrt{\frac{D^2}{2}}$$

A cutoff of $C/2$ was then applied for ASm⁶A identification in individual F1i or F1r samples, considering the bidirectional nature of the fluctuation. These analyses were separately conducted on C57-specific and PWK-specific adenine sites. The final cutoff (0.6) was determined by averaging the results obtained from both datasets.

### Reproducibility between paired F1i and F1r samples

Candidate ASm⁶As in each sample were identified with criteria: $P < 0.05$, FDR $< 0.1$ and $\left|\log_2(cpFC)\right| > 0.6$. Candidates present in both F1i and F1r samples within each F1i-F1r group were identified as ASm⁶A sites. These sites were further categorized into two major classes based on allelic bias directions in paired F1i and F1r samples: seq-ASm⁶A and parent-ASm⁶A. Each category was sub-classified into two classes:

C57-biased seq-ASm⁶A: $F1i\log_2(cpFC) > 0.6$, $F1r\log_2(cpFC) > 0.6$;
PWK-biased seq-ASm⁶A: $F1i\log_2(cpFC) < -0.6$, $F1r\log_2(cpFC) < -0.6$;
Mat-biased parent-ASm⁶A: $F1i\log_2(cpFC) < -0.6$, $F1r\log_2(cpFC) > 0.6$;
Pat-biased parent-ASm⁶A: $F1i\log_2(cpFC) > 0.6$, $F1r\log_2(cpFC) < -0.6$.

To obtain highly reproducible ASm⁶As, we further filtered the identified ASm⁶A sites in each F1i-F1r group. We set a criterion that an ASm⁶A site must be allelically detectable in at least four samples from the same tissue. Additionally, its group-level $\log_2(cpFC)$ or $\log_2(mpFC)$ must align directionally with the corresponding tissue-level $\log_2(cpFC)$ or $\log_2(mpFC)$ for F1i and F1r samples, respectively.

### Quantification of SNP distribution around m⁶A sites

To investigate the association between SNP distribution and allelic m⁶A imbalance levels of seq-ASm⁶A, we introduced a weighted scoring metric. For a given m⁶A site, we assessed two key features: the density and distance of SNPs located within a 200-nt region (from 100 nt upstream to 100 nt downstream of the m⁶A). This region was divided into 20 10-nt bins, numbered from 1 to 10 based on their distance from the m⁶A site. The two nearest bins on both sides of the m⁶A were numbered 1, while the two farthest bins were assigned 10. The SNP distribution score ($S$) was calculated as

follows:

$$S = \sum_{n=1}^{N} \frac{1}{D_n} + \sum_{n=1}^{N} \frac{1}{B_n}$$

Here, $N$ represents the total count of SNPs in the 200-nt region, $D_n$ denotes the distance of each SNP, and $B_n$ is the bin number of each SNP. A higher score indicates greater SNP density in closer regions to the analyzed m⁶A site.

### Enrichment analysis of SNP locations flanking seq-ASm⁶A sites

The nearest SNP locations to testable m⁶A sites within their flanking 100-nt regions were employed for the enrichment analysis. Positions within the 100-nt flanking region of m⁶A were numerically labeled with respect to the transcript strand. For each position, two counts were obtained:

$S_i$: count of highly reproducible seq-ASm⁶A sites with the nearest SNP located at this position;

$T_i$: count of testable m⁶A sites with the nearest SNP located at this position.

Here, $i$ represents the label for each position ($[-100, -1]$, $[1, 100]$). The union of seq-ASm⁶A sites from all F1i-F1r groups and the union of testable m⁶A sites from all F1i-F1r groups were considered in the counting process. $S_i$ was treated as a binomial random variable:

$$S_i \sim Binomial(n = T_i, p = R)$$

where $R$ was calculated based on the average $S_i/T_i$ within the 200-nt region:

$$R = \frac{1}{200}\left(\sum_{i=-100}^{-1} \frac{S_i}{T_i} + \sum_{i=1}^{100} \frac{S_i}{T_i}\right)$$

Positions with $P < 0.05$ in the binomial test were identified as candidate *cis*-regulatory sites associated with m⁶A methylation and allelic m⁶A imbalance.

### Allelic methylation imbalances for m⁶A sites with motif variations

In each tissue, allelically detectable m⁶A sites with two distinct allelic motif sequences were identified, based on the intersection of the motif regions with high-confidence SNPs using bedtools (v.2.26.0) (Quinlan, 2014; Quinlan and Hall, 2010). For these sites, the metric $\log_2(cpFC)$ was employed to quantify the methylation difference between C57 and PWK motif sequences. Specifically, a $\log_2(cpFC) > 0$ implies that the methylation level tends to decrease when the C57 motif sequence mutates to the PWK motif sequence. Conversely, a $\log_2(cpFC) < 0$ indicates an opposite directional change. Based on the identified high-confidence m⁶A sites, we ranked motif sequences by their occurrence frequencies in each tissue, and labeled the top 16 sequences with their corresponding sequences while grouping all other sequences as "Other." Then, we conducted a pairwise comparative analysis involving these

17 sequences, wherein we assessed the methylation differences between each pair. This assessment was based on $\log_2(cpFC)$ values of m⁶A sites with corresponding allelic motif sequences. For each pair of sequences, the average $\log_2(cpFC)$ was computed and utilized for visualization through heatmaps, which systematically illustrated the impact of motif variations on allelic m⁶A methylation levels. Heatmaps were generated by the R package pheatmap v.1.0.12 (https://CRAN.R-project.org/package=pheatmap).

## Quantification of motif variation effects on m⁶A methylation

For m⁶A sites with SNP located at the motif positions (the two alleles possessed distinct motif sequences), we utilized motifbreakR (v.2.10.0) (Coetzee et al, 2015) to quantify variation (motif-overlapping SNP) effects on m⁶A methylation. This analysis was based on motif enrichment analysis encompassing all high-confidence m⁶A sites, which was conducted using the findMotifsGenome.pl module under HOMER (v.4.11) (Heinz et al, 2010). For each m⁶A site with motif variations, motifbreakR evaluated the impact of SNPs on m⁶A methylation when the motif sequence shifts from C57 to PWK motif sequences. Specifically, positive values indicate a stronger match of the PWK motif with the HOMER-derived consensus motif, while negative values indicate a stronger match of the C57 motif with the consensus motif.

## Identification of m⁶A sites exhibiting ASE

For each m⁶A site, we evaluated allelic expression imbalance within F1i-F1r groups using the defined ARR metrics ($ARR_{c57,input}$ and $ARR_{mat,input}$). An m⁶A site exhibiting sequence-dependent ASE was identified when its ARR values favored the same strain in both F1i and F1r samples, resulting in two sub-classes:

C57-biased expression: $ARR_{c57,input} > 0.6$ in both F1i and F1r samples.

PWK-biased expression: $ARR_{c57,input} < 0.4$ in both F1i and F1r samples.

An m⁶A site exhibiting parent-of-origin-dependent ASE was identified when its ARR values favored the same parent in both F1i and F1r samples, leading to two sub-classes:

Mat-biased expression: $ARR_{mat,input} > 0.6$ in both F1i and F1r samples.

Pat-biased expression: $ARR_{mat,input} < 0.4$ in both F1i and F1r samples.

Balanced expression was also defined: $0.4 \leq ARR_{c57,input} \leq 0.6$ in both F1i and F1r samples.

## Collection of known imprinted genes

The known imprinted genes were collected from relevant reports (Crowley et al, 2015; Gregg et al, 2010; Perez et al, 2015; Tucci et al, 2019), and databases (https://www.geneimprint.com/site/genes-by-species.Mus+musculus).

# Data availability

The sequence data generated in this study have been deposited in the NCBI Gene Expression Omnibus (GEO), under accession number GSE265979.

The source data of this paper are collected in the following database record: biostudies:S-SCDT-10_1038-S44318-025-00476-3.

# Peer review information

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

## Acknowledgements

We thank XS Xiong and J Xu for helpful discussion. This work was supported by the Ministry of Science and Technology of China (2022YFC3400400 and 2022YFA0912900), National Natural Science Foundation of China (32425034, 92253202, 32271499, 32270644, and 32100461), Pearl River Talent Recruitment Program (2019ZT08Y485), and Shenzhen Bay Scholars Program.

## Author contributions

**Ying Zhang**: Conceptualization; Data curation; Formal analysis; Validation; Investigation; Visualization; Methodology; Writing—original draft; Writing—review and editing. **Ze-Yu Zhang**: Resources; Validation. **Hong-Xuan Chen**: Data curation; Validation; Investigation; Methodology. **Chao Liu**: Resources; Data curation. **Biao-Di Liu**: Data curation; Methodology. **Ye-Lin Lan**: Data curation; Formal analysis; Investigation; Methodology. **Ying-Yuan Xie**: Data curation; Investigation; Methodology. **Tao Chen**: Validation; Methodology. **Shaobo Chen**: Data curation; Validation. **Guihai Feng**: Supervision. **Zhang Zhang**: Funding acquisition; Writing—review and editing. **Wei Li**: Supervision. **Nan Cao**: Supervision; Writing—review and editing. **Xiu-Jie Wang**: Supervision; Project administration. **Guan-Zheng Luo**: Conceptualization; Supervision; Funding acquisition; Project administration; Writing—review and editing.

Source data underlying figure panels in this paper may have individual authorship assigned. Where available, figure panel/source data authorship is listed in the following database record: biostudies:S-SCDT-10_1038-S44318-025-00476-3.

## Disclosure and competing interests statement

The authors declare no competing interests.

# Expanded View Figures

**Figure EV1.   Allele-specific analysis of m⁶A methylation profiles.**

(**A**) Workflow for upstream data analysis to identify ASm⁶A sites. (**B–D**, **F**) Metagene profiles showing the distributions of m⁶A peak summits (**B**), GLORI-detected m⁶A sites (**C**), high-confidence m⁶A sites (**D**) and allelically detectable m⁶A sites (**F**). Motif analysis of high-confidence m⁶A sites is shown in (**D**). (**E**) Upset plot illustrating intersections of high-confidence m⁶A sites across all samples. The top 30 intersections, ranked by size, are displayed. The deep blue highlights the intersection of all samples. The pie chart shows the distribution of m⁶A sites identified in all 14 samples, 3–13 samples, and fewer than 3 samples. (**G**) PCA of allele-specific m⁶A levels in 12 cerebellum sub-samples. Each dot represents an allele-specific sub-sample, with shape denoting sex (circle for female, triangle for male) and size denoting age (small for P0, large for P7). Dot color indicates genotype, with outline color denoting parent-of-origin and fill color denoting strain. The first two PCs are shown. Further components are shown in (**H**). PC, principal component. (**H**) Scree plots presenting the percentage of explained variances for identified components in PCA analyses. Color scheme for samples depicted in (**D–F**) is illustrated in the legend of (**B**).

▶

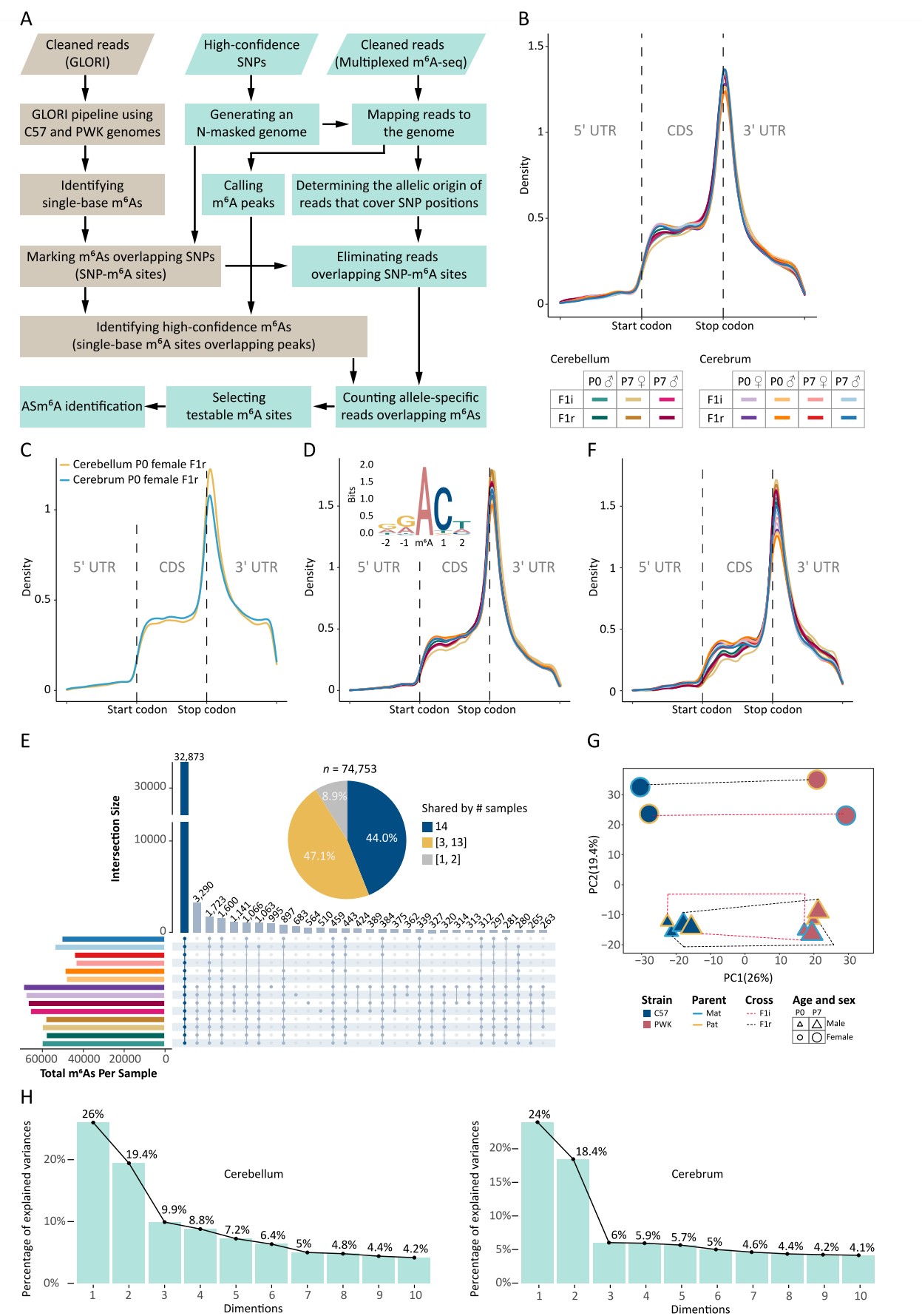

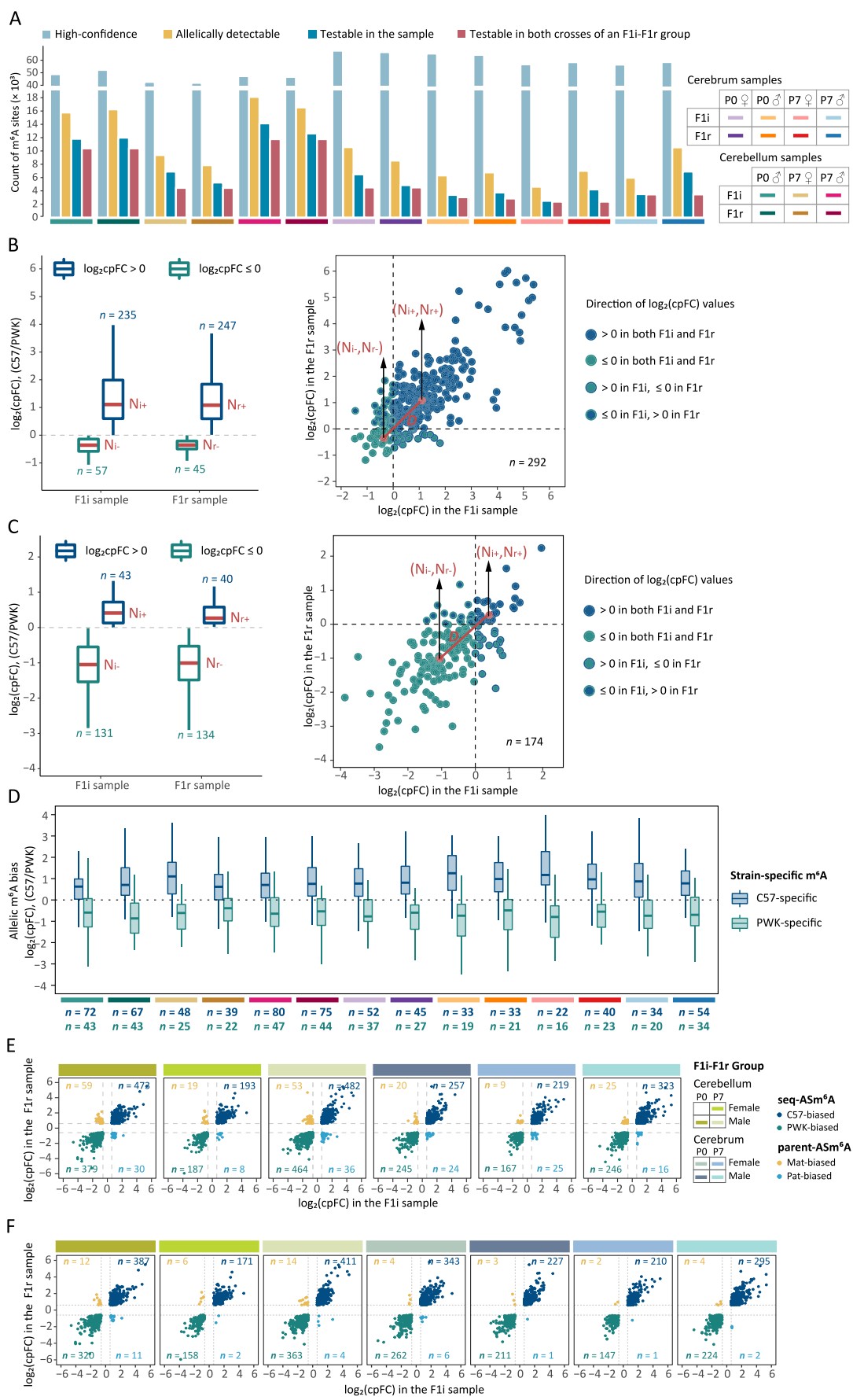

**Figure EV2.  Identification of candidate ASm⁶A sites.**

(A) Bar plots displaying counts of m⁶A sites categorized into four groups (see "Methods"). ASm⁶A identification within each F1i-F1r group utilized sites testable in both crosses. (B, C) Evaluation of the fluctuation in $\log_2(cpFC)$ values of positive controls. Positive controls encompass both C57-specific (B) and PWK-specific (C) m⁶A sites. Box plots depict the distribution of positive and negative $\log_2(cpFC)$ values within F1i and F1r samples, resulting in four datasets whose medians are used to evaluate the fluctuation (see "Methods"). $N_{i+}$ and $N_{r+}$ denote the medians of positive $\log_2(cpFC)$ values in F1i and in F1r samples, respectively. Similarly, $N_{i-}$ and $N_{r-}$ denote the medians of negative $\log_2(cpFC)$ values in F1i and in F1r samples, respectively. $D$ represents the overall fluctuation level of $\log_2(cpFC)$ values in the F1i-F1r group, which was assessed using Euclidean distance (see "Methods"). (D) Box plot illustrating allelic m⁶A difference for positive controls across all m⁶A-seq samples. Color scheme for the samples is identical to that of (A). (E) Four-quadrant scatter plots illustrating ASm⁶A distribution across six F1i-F1r groups. (F) Four-quadrant scatter plots showing highly reproducible ASm⁶A sites. These sites were chosen from ASm⁶A sites within each F1i-F1r group based on two criteria: each ASm⁶A site must be detectable in ≥ 4 samples, and its group-level allelic bias must align directionally with the tissue-level allelic bias (see "Methods"). The color scheme is consistent with that of (E). (B–D) The top, middle, and bottom lines of the box represent the upper quartile (Q3), median, and lower quartile (Q1), respectively. The upper whisker extends to the maximum value provided it is not larger than $(Q3 + 1.5 \times IQR)$ (where $IQR = Q3 - Q1$), while the lower whisker extends to the minimum value provided it is not smaller than $(Q1 - 1.5 \times IQR)$. Data points beyond the whiskers are considered outliers and are not displayed. Source data are available online for this figure.

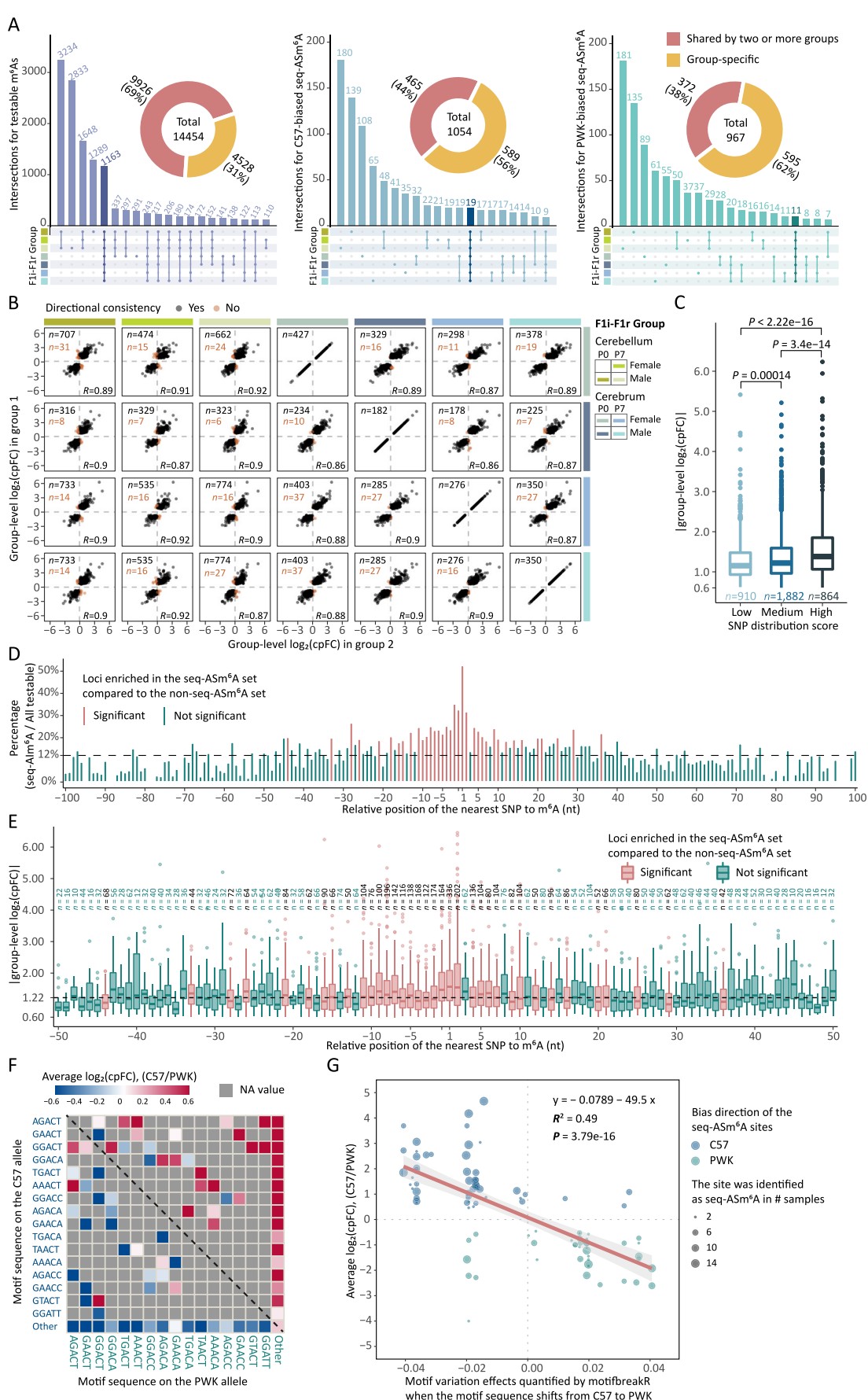

◄   **Figure EV3.   *Cis*-regulatory effects on allelic m⁶A levels.**

(**A**) Upset plots illustrating intersections of testable m⁶A sites (left), C57-biased seq-ASm⁶A (middle) and PWK-biased seq-ASm⁶A (right) across all F1i-F1r groups. The top 20 intersections, ranked by size, are displayed. The deeper color within each plot highlights the intersection of all groups. Donut charts adjacent to each plot show the proportions of group-specific and group-shared sites. (**B**) Correlation analysis of allelic bias in seq-ASm⁶A methylation across F1i-F1r groups. Each scatterplot depicts the intersection between seq-ASm⁶A sites in a cerebrum group (Group 1) and allelically detectable m⁶A sites in another group (Group 2). Pearson's $R$ and the count of m⁶A sites are annotated. (**C**) Box plot showing the allelic imbalance levels of seq-ASm⁶A sites categorized by SNP distribution scores (see "Methods"). Statistical analysis utilized the two-sided Wilcoxon rank-sum test. (**D**) Enrichment analysis of SNP positions in the flanking 100 nt region of seq-ASm⁶A sites (see "Methods"). Each bar indicates the ratio of seq-ASm⁶A sites to all testable m⁶A sites with their nearest SNP at the position. The dashed line indicates the average ratio. Statistical analysis employed a one-sided Binomial test ($*P < 0.05$, $**P < 0.01$, $***P < 0.0001$). (**E**) Box plots illustrating the distribution of allelic m⁶A imbalance levels for seq-ASm⁶As with the nearest SNP at specific positions. For each position, the count ($n$) of seq-ASm⁶A sites with their nearest SNP located at that position is labeled. (**F**) Heatmap depicting allelic m⁶A differences among motif pairs in the cerebrum (see "Methods"). Motifs are ranked by occurrence frequency within all high-confidence m⁶A sites in the cerebrum. For each motif pair, color represents the average $\log_2(cpFC)$ value of m⁶A sites with corresponding motif variations. Blue and red indicate higher m⁶A levels on the PWK and C57 alleles, respectively. Gray represents NA values. (**G**) Linear regression analysis of motif variation effects and allelic m⁶A differences in cerebral seq-ASm⁶As. The variation effects were determined using motifbreakR (see "Methods"). Shaded areas represent 95% confidence intervals of the regression line. Statistical significance was assessed using the Student's $t$ test. (**D, E**) SNP positions are labeled with respect to the transcript strand. (**C, E**) The top, middle, and bottom lines of the box represent the upper quartile (Q3), median, and lower quartile (Q1), respectively. The upper whisker extends to the maximum value provided it is not larger than $(Q3 + 1.5 \times IQR)$ (where $IQR = Q3 - Q1$), while the lower whisker extends to the minimum value provided it is not smaller than $(Q1 - 1.5 \times IQR)$. Data points beyond the whiskers are considered outliers and are plotted individually.

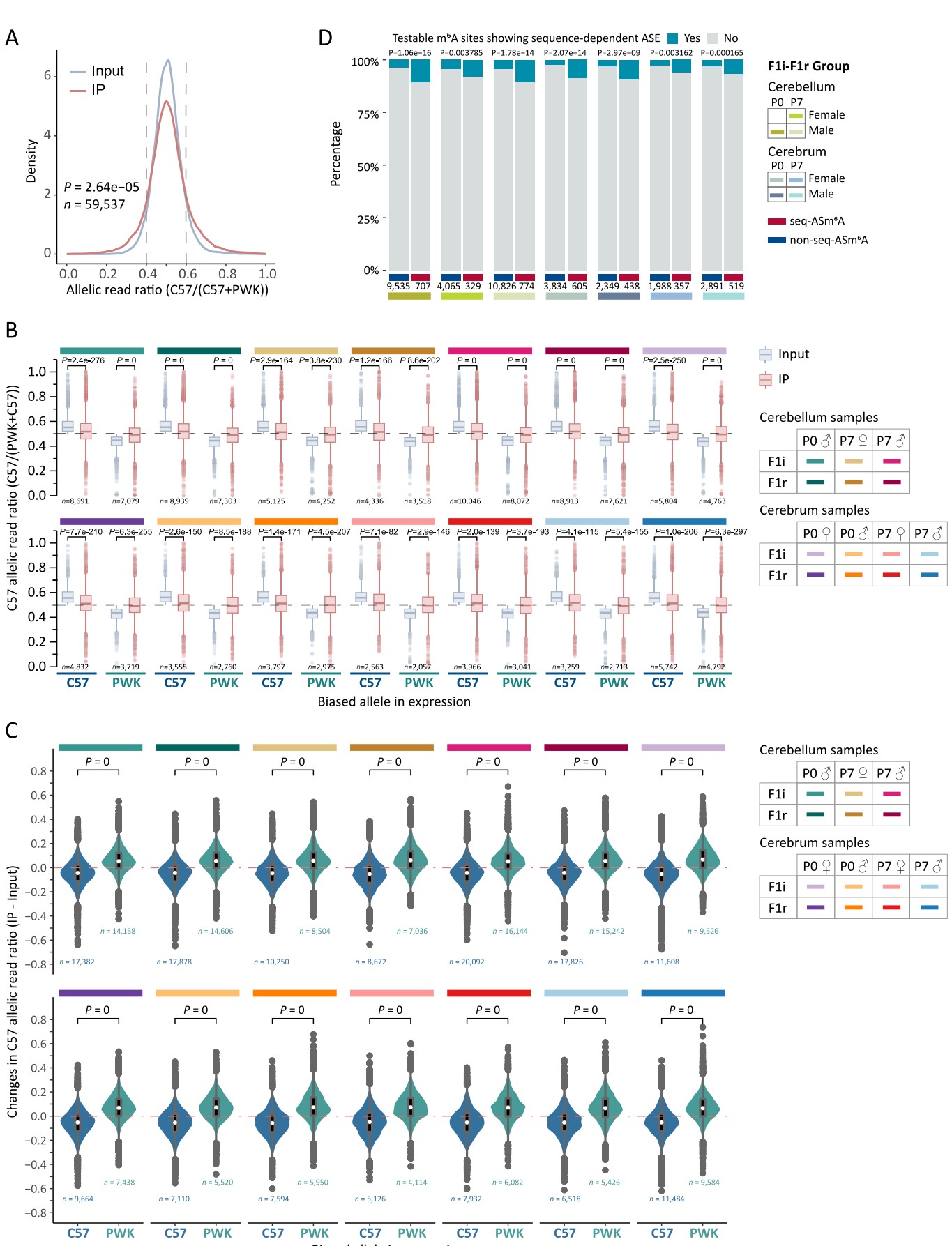

**Figure EV4.   Opposing allelic preferences between sequence-dependent m⁶A methylation and expression.**

(A) Density plot illustrating allelic read ratio distributions in untreated input and IP samples for all allelically detectable m⁶A sites. Statistical significance was assessed using the two-tailed paired Student's $t$ test. (B) Box plots comparing C57 allelic read ratios between untreated input and IP samples. Statistical analysis utilized the two-sided paired Wilcoxon rank-sum test. ***$P < 0.0001$. (C) Distributions of differences in C57 allelic read ratios between untreated input and IP samples. White circle indicates median, and violin-shaped areas depict kernel density estimates of data distribution. Statistical analysis employed the two-sided Wilcoxon rank-sum test. (D) Stacked bar charts showing the proportions of testable m⁶A sites with ASE and without ASE. Comparison was conducted between seq-ASm⁶A and non-seq-ASm⁶A sites using the two-sided Pearson's chi-squared test. The bottom numbers indicate the number of m⁶A sites. (B, C) The top, middle, and bottom lines of the box represent the upper quartile (Q3), median, and lower quartile (Q1), respectively. The upper whisker extends to the maximum value provided it is not larger than $(Q3 + 1.5 \times IQR)$ (where $IQR = Q3 - Q1$), while the lower whisker extends to the minimum value provided it is not smaller than $(Q1 - 1.5 \times IQR)$. Data points beyond the whiskers are considered outliers and are plotted individually. The count ($n$) of m⁶A sites for each plot is labeled in the figure.

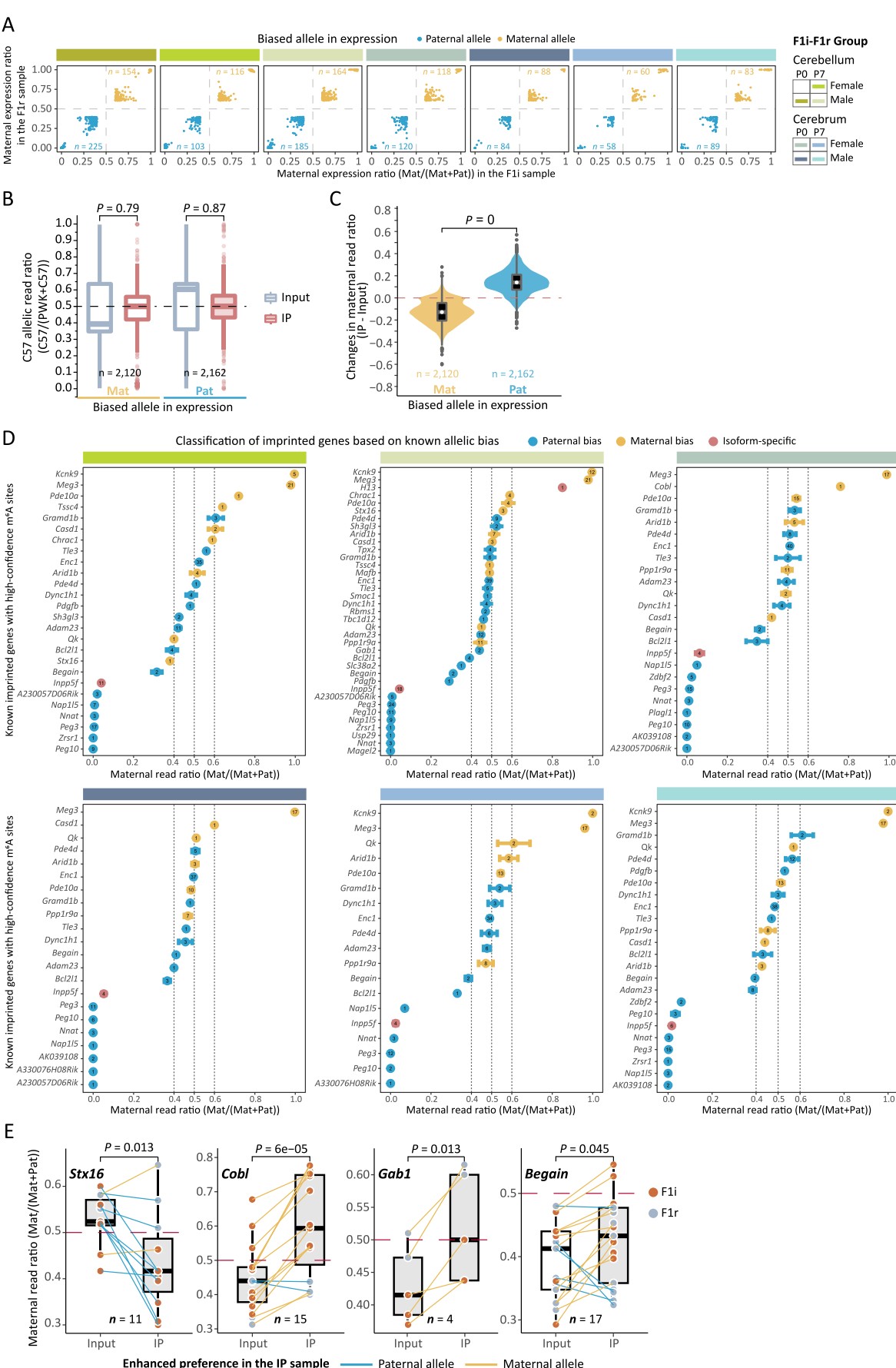

**Figure EV5.   Parental effects on m⁶A methylation and expression.**

(**A**) Scatter plots showing m⁶A sites exhibiting parent-of-origin-dependent ASE in each F1i-F1r group. These sites were identified based on maternal read ratio in untreated input samples (see "Methods"). (**B**) Box plots comparing C57 allelic read ratios between untreated input and IP samples, with m⁶A sites exhibiting maternal ($n = 2120$) and paternal ($n = 2162$) allelic expression biases shown separately. Statistical analysis utilized the two-sided paired Wilcoxon rank-sum test. (**C**) Distributions of differences in maternal allelic read ratios between untreated input and IP samples. Statistical analysis employed the two-sided Wilcoxon rank-sum test. White circle indicates median, and violin-shaped areas depict kernel density estimates of data distribution. (**D**) Imprinted genes harboring high-confidence m⁶A sites in each F1i-F1r group. Each point represents a gene, color-coded by its reported imprinted category and labeled with the number of m⁶A sites. Maternal expression ratio is shown as mean ± standard error across all m⁶A sites within each gene. The genes are ranked by maternal read ratio. The dotted lines represent the cutoffs for identifying m⁶A sites showing parent-of-origin-dependent ASE (see "Methods"). Color scheme for groups is provided in (**A**). (**E**) Box plots illustrating differences in maternal allelic read ratios between untreated input and IP samples for allelically detectable m⁶A sites within known imprinted genes. Four representative genes are shown, with the count ($n$) of m⁶A sites labeled in the plot. Sites are labeled by cross (F1i or F1r) to distinguish sequence- and parent-of-origin-dependent effects. Statistical significance was assessed using the two-tailed paired Student's *t* test. (**B, C, E**) The top, middle, and bottom lines of the box represent the upper quartile (Q3), median, and lower quartile (Q1), respectively. The upper whisker extends to the maximum value provided it is not larger than ($Q3 + 1.5 \times IQR$) (where $IQR = Q3 - Q1$), while the lower whisker extends to the minimum value provided it is not smaller than ($Q1 - 1.5 \times IQR$). Data points beyond the whiskers are considered outliers and are plotted individually.

