## [Peer Review File · The EMBO Journal]

Sequence and parent-of-origin dependent m6A contribute to allele-specific gene expression

Ying Zhang, Zeyu Zhang, Hongxuan Chen, Chao Liu, Biaodi Liu, Ye-Lin Lan, Yingyuan Xie, Tao Chen, Shaobo Chen, Guihai Feng, Zhang Zhang, Wei Li, Nan Cao, Xiu-Jie Wang, and Guanzheng Luo

Corresponding author(s): Guanzheng Luo (luogzh5@mail.sysu.edu.cn)

Review Timeline:

Submission Date:	24th Jul 24
Editorial Decision:	28th Oct 24
Revision Received:	20th Feb 25
Editorial Decision:	31st Mar 25
Revision Received:	13th Apr 25
Accepted:	29th Apr 25

Editor: *Cornelius Schneider*

Transaction Report:

Dear Dr. Luo,

Thank you for submitting your manuscript for consideration by the EMBO Journal. It has now been seen by three referees whose comments are shown below.

As you can see all referees think that the results are of interest and that the experiments and data are of high quality. However, referee #1 remarks that the manuscript would benefit from improved data representation and referee #3 finds that the results would need to be put better into context especially regarding the modification states of the parental strains (please also see the referee cross-commenting by referee #3). All together we think that the referee comments are reasonable and productive and given the referees' positive recommendations, I would like to invite you to submit a revised version of the manuscript, addressing the comments of all three reviewers. I should add that it is EMBO Journal policy to allow only a single round of revision, and acceptance of your manuscript will therefore depend on the completeness of your responses in this revised version.

Thank you for the opportunity to consider your work for publication. I look forward to your revision.

Yours sincerely,

Cornelius Schneider, PhD
Editor
The EMBO Journal
c.schneider@embojournal.org

We realize that it is difficult to revise to a specific deadline. In the interest of protecting the conceptual advance provided by the work, we recommend a revision within 3 months (26th Jan 2025). Please discuss the revision progress ahead of this time with the editor if you require more time to complete the revisions. Use the link below to submit your revision:

Referee #1:

In their manuscript "Sequence and parent-of-origin dependent m6A contribute to allele-specific gene expression" Zhang et al. analyse m6A mRNA modifications and corresponding gene expression data to investigate allele-specific biases driven by sequence difference and/or parental origin. The authors posit that m6A modifications may contribute to allele-specific gene expression post-transcriptionally by driving allele-specific mRNA decay (and through other unspecified mechanisms). In principle, this is an interesting and comprehensive analysis of a nice set of data, but as it stands, the paper is very difficult to read and understand. The results and figures can only be understood by reading the methods section in parallel and very carefully. The authors need to define much more explicitly what is being compared in their different analyses. Also, the figures are very busy; more is not always better. Some panels show the same data just in a slightly different way. It would make the paper easier to digest if the figures were slimmed down.

For any future revision, please do not separate figure legends from the figures and add line numbers. Here I will refer to page numbers and figures for detailed comments and questions.

Page 2 Abstract: The abstract needs to mention what m6A is. It doesn't say that it's an mRNA modification.

Fig 1a: The diagram does not help to understand what the samples are. It would be better to have supplementary table 1 as a table in the main part of the paper. Especially the middle table in the diagram is very confusing. It's not clear how you get seven and nine samples from what looks like 2x4 age- and sex-matched offspring from reciprocal crosses.

Fig 1d: Here it would help to connect, for example, the two sub-samples for one of the P0 male samples with a dotted line and to explain in the text that these come from the same physical sample.

Page 7: It's confusing to call a pair of samples 'group' because group implies a number higher than 2. Make it very explicit in the text, e.g. say "The two age- and sex-matched F1i and F1r samples are from here on referred to as a group".

Page 8: If the A in the positive control m6As is a SNP how can the bias only be 80%? If this is a SNP, by definition the other allele has a base other than an A so can't have m6A.

Fig 2a, b: I don't think these need to be in the main figure. Also, please define all abbreviations and labels in the figure legends. I had to dig through the methods to find what Ni⁺ etc are.

Fig. 2e. The legend for the F1i-F1r group needs to be included here.

Page 10: Here the sentence "We then systematically assessed ASm6As across F1i-F1r groups" means ASm6A analysis WITHIN groups. On page 11 the authors write practically the same, i.e. "we next asked whether (...) seq-m6As show consistency across F1i-F1r groups", to mean BETWEEN groups. Given the complexity of this analysis it is paramount that the authors are very clear on what they mean.

Page 10: '69% of testable sites exhibited group-sharing, which decreased to 41% for seq-m6As'. What does this mean? How are the testable sites different? Is the rest parent-driven?

Fig 3g: It's not clear from the heatmap in what order the motifs are ranked. Also, setting NA values to zero is misleading. A better option would be to have a separate colour (a light grey) for the NA values.

Fig 3h doesn't add much, it's just another depiction of the heatmap results. It's also not explained in the figure legend what the red numbers along the x-axis are.

Fig 3i: That correlation doesn't look great, weak R2. The data fall mostly into four distinct clusters along the x-axis.

Page 15: "SNP effects on m6A consensus motif". From the text and figure panel it's not at all clear what is meant by that. Only after reading the methods can this be understood.

Figure 4e: What are the numbers along the x-axis? Number of seq-ASm6A sites?

Page 19: Here the wording needs to be made much clearer. "allelic biases in methylation (...) displayed a trend of favoring the parental allele with lower expression level" is very confusing. "Higher m6A levels correlated with lower gene expression levels" is probably meant here. Similarly "changes in allelic bias from m6A expression to methylation." What does this mean? Or "shifts in allelic bias towards the opposing allele" implies a measurement before and after methylation which is not the case. Overall this paragraph needs more precise phrasing.

Fig 5b: Not sure what this adds beyond 5a.

Fig 5c. Are the dotted lines cut-offs for allele bias? Make clear in figure legend.

Fig 5e. If ASE is present, one would expect the whole gene to be biased in the same direction. So not sure what is gained by looking at individual m6A sites across the gene.

Page 21: "opposing" rather than "contradictory"

Page 24: "we quantified the allelic imbalance shift from expression to methylation". What does this mean? As above, "shift" implies a before and after, and a shift from expression to methylation doesn't make sense.

The authors state they propose two mechanisms, but the second is just a vague allusion to m6A reader binding. A bit more detail would be good.

Referee #2:

In this work the authors have mapped thousands of allele-specific m6A (ASm6A) sites with statistically significant. They show that over 90% of these sites are not due to motif sequence variations but most likely cis sequence changes within 50 nt sliding window. These allele-specific m6A sites mostly affect transcript decay, consistent with the known roles of m6A. The overall study design and execution are good, with valuable information provided. I think this is a good contribution to the field and the results may stimulate future investigations. I have a few comments below:

1. Among thousands of sites identified if they used GLORI, how do they control potential deamination variations in GLORI. Is that a problem for their work?
2. Could the authors speculate what are cis-regulatory factors for allele-specific m6A sites beyond the motif sequence? I assume many of these are allele-specific sequence changes that lead to altered RBP binding? Any specific motifs coming out of their analyses? It is hard to think all effects from these RBP binding to the target transcript. Whatelse?
3. What function categories that may enrich parent-of-origin effect of allele-specific m6A sites?

Referee #3:

This was a study of the post-transcriptional mRNA modification N6-methyladenosine (m6A) in two brain tissues of F1 hybrids between two divergent mouse strains. The main experiment used m6A-seq on 14 samples, using a method which relies on immunoprecipitation of fragmented barcoded pooled mRNA with anti m6A antibody. This results in sequencing only those mRNA fragments that contained m6A, and these were then also compared with non-immunoprecipitated fragmented barcoded pooled mRNA (the "input sample") to assess the overall abundance of transcripts independent of m6A. The second experiment used glyoxal and nitrate-mediated deamination (GLORI) treatment on two samples. This treatment deaminates only non-methylated adenosines to inosines, which can then be identified by comparing the resulting RNA-seq with reference genomes. By overlapping the datasets from these two different methods, high confidence m6A sites could be identified. The rest of the manuscript concerned an in depth bioinformatic analysis of these datasets to detect tissue, sex, age effects on m6A and strain and parent of origin effects on the allelic methylation and expression of transcripts as well as identifying cis-regulatory variants enriched in regions flanking m6A sites, making use of reciprocal crosses and paired sample design.

The major findings of the study were the identification of cis-regulatory variants in the 50 nucleotides to the 5' and 3' of m6A sites, and the detection of sequence dependent, and parent of origin dependent allele specific transcript methylation. It was particularly striking that there was a negative relationship between the allele specific methylation and allele specific expression, hinting at the possible role of allele-specific m6A in downregulating transcript abundance.

In mammalian cells m6A is the most abundant post-transcriptional mRNA modification and has been explored in many previous studies. The novelty in this study is the comparison of allele-specific expression with allele-specific methylation (adding the more robust method detecting m6A with both the competitive immunoprecipitation on pooled barcoded mRNA and cross comparison with GLORI seq), as well as the elucidation of sequence-dependent and parent of origin dependent effects. These novel finding presented in this manuscript therefore represent an important advance and will be of general interest and provide useful starting points for future research on the post transcriptional determinants of allele-specific expression. I do however have some reservations regarding the study.

My major reservation is that although an in-depth study and analysis of m6A has been made for the F1 hybrids, the study provides no experiment or information on m6A in the equivalent tissues (cerebrum and cerebellum) and ages (0 and 7 days after birth) of the two parental mouse strains, C57 and PWK. Are they the same sites as seen in the C57 and PWK alleles in the F1? Without any information on the situation or different phenotypes of the parental strains it is hard to understand what is the biological relevance of the allele specific m6A and ASE in the F1 hybrids. On this theme one key study on transcriptome-wide m6A not cited or discussed is Xu et al 2021 <https://doi.org/10.3389/fpls.2021.685189>, which studied transcriptome-wide m6A in Arabidopsis strains and their hybrids. In this study Xu et al observed a disappearance of m6A differences at allelic sites in hybrids, with allelic bias in m6A in hybrids being very rare. This seems like a potentially large biological difference between the use of m6A in plants and animals that needs further investigation.

My other reservations are more minor. As line numbers do not appear in the manuscript I received I have used page numbers and section headings to refer to the locations in the manuscript described below.

1. On p39 of the Methods (section "ASm6A identification and classification") it is stated that a binomial model was employed to assess the statistical significance of allelic methylation imbalance. Was this read count data assessed for overdispersion, which is a very common feature of this type of data (Castel et al. Genome Biology (2015) 16:195 DOI 10.1186/s13059-015-0762-6). If the data was over dispersed then this could have resulted in an overinflation of the significance levels calculated using a binomial model. The data should be tested for distribution and if the data is indeed over dispersed, a beta binomial or some other more appropriate model should be used instead.
2. In the introduction and at multiple points throughout the results and discussion the mechanism of m6A exerting its function by promoting mRNA degradation is mentioned. On p16 in the results section it is mentioned that "widespread shifts in allelic bias towards the opposing allele in m6A methylation were observed" - which I take as meaning that the allele with higher overall expression tends to have lower m6A methylation. I find it very hard to assess on the data in the manuscript alone which is a measurement of the steady state expression levels and m6A methylation at a snapshot in time whether this conclusion is justified in this context. This is because there is no attempt to quantify transcript degradation in this manuscript. The opposing directions of the expression and methylation signals do not imply causation - it could just be that linked SNPs that are responsible both for cis-regulation of expression and for methylation. To make such a certain statement of the mechanism would require testing this hypothesis by for example measuring transcript abundance over time while either blocking or not blocking m6A methylation, or employing gene editing techniques to swap m6A motif sites between the two strain genetic backgrounds. The proposal on p18 that "Based on these findings, we propose that seq-ASm6As primarily participate in regulating ASE by promoting allele-specific mRNA degradation" should therefore be less overstated.
3. I spent a lot of time wondering why were cerebrum and cerebellum selected, and why postnatal day 0 and day 7? The explanation for the choice of tissue is hidden away in the methods section (under subheading "Multiplexed m6A-seq), and there is no explanation for the choice of ages. This should be included in the introduction or results.
4. Frequent mention of age matched pairs of samples from the reciprocal crosses was made, but to make sense of the pairs it was necessary to look very carefully at TableS1, as figure 1a was not easy to interpret. Although the total number of samples sequenced was 16, there was only one matched pair for each combination of conditions of sex, age and tissue.
5. IP (m6A immunoprecipitation) is first mentioned in manuscript the results section on p16 in this sentence "To address this, we calculated paired allelic read ratios (ARRs) for both input (expression) and IP (methylation) samples at each m6A site, *RRc57,input* and *ARRc57,ip*, quantifying sequence-dependent allelic preferences (see Methods)." The explanation of what "IP" stands for is not given until the methods section on p31: "Multiplexed m6A immunoprecipitation was conducted 2 rounds, based on published work". The multiplexing is an important part of this protocol to make the m6A detection comparable between samples and more explanation of this is needed in the methods, not requiring all readers to go and read Dierks et al 2021 methods paper in full. The explanation for what IP/IP samples mean should also be brought forwards to the results section. I found this particularly confusing as the IP samples were compared with input samples, which also sounds a bit similar to IP. This would be less confusing if the more meaningful label of "expression samples" and "methylation samples" as used on page 16 was used throughout, or adding "untreated" before "input".
6. P0 and P7 are not explained until the methods section and yet are mentioned in the results and figures. Non mouse biologists will not be familiar with these terms. Explain them at the first mention.
7. Figures: A lot of the figures were unnecessarily large and complicated to read. This was compounded by the choice of extremely similar colours, for example dark blue and teal green in Figure 1d. Some panels e.g. 1e, 2e, are showing very similar trends across the samples and could be replaced with either the average or an example and the explanation that the other samples followed the same trend. 3b did not require a plot and could be replaced with a single sentence in the text. 5c has no explanation for the ordering on the vertical axis - I presume the genes are ranked by maternal read ratio? Supplementary figure 2a there no need for a discontinuity in the graph, simply make the plot taller, it's supplementary so space constraint not an issue. Also in supplementary figure 2a the groups of m6A sites are in almost indistinguishable shades of blue. Figure 5f had an excessive number of symbols and colours in the key making it virtually unreadable. It would have been as easy to just label the data points with text.
8. Punctuation on the same line as equations in the methods section leads to confusion. There is unnecessary and confusing punctuation after every equation e.g. p34 comma after equation for *Lij*, on p36 full stop after equation for $\log_2(cpFC)$. Something strange is going on with brackets on p38 in the methods section (mixture of normal and square). It was unclear to me if the bracket shapes have any significance.
9. On p26 in the legend of Figure 1c it is unclear what the n=2153 referring to?
10. On p37 in the methods section mention is made in the "Inter-sample same-strain and intra-sample different-strain allelic m6A imbalance to "negative control datasets". This was inadequately explained what it is controlling for - is this trying to work out the background level of allelic methylation?
11. There was a bit of confusion between where 100 nucleotide and 50 nucleotide regions flanking m6A sites were being considered between the text and the figures. If I understood correctly, 100 nucleotide flanking regions were first considered but all identified potential cis-regulatory sites fell within the 50 nucleotide flanking regions? This should be made clearer in the text.

Referee crosscommenting:

Reviewer 3 here. I just want to take this opportunity to further explain my comment regarding the lack of information on gene expression and m6A in the parental strains. My comment was that it is difficult to understand the biological significance of the

allele-specific ASE and m6A in the F1 without information regarding m6A in the parental strains. I did not necessarily expect this to be addressed by the generation of the (large) missing datasets as I understand this would be time consuming if there is no data available. I think a good discussion on this point citing the literature regarding the two strains and discussing the relevance of cerebrum and cerebellum gene expression and posttranscriptional modifications to mouse biology would be sufficient. So perhaps we are not in as much disagreement as it seems.

Point-by-point response to reviewer comments

Summary

We thank the editors and reviewers for their feedback and insightful comments, which have significantly helped us to improve the manuscript. We revised the manuscript and updated figures accordingly. We hope that all the concerns and questions raised by the reviewers have now been fully-addressed. Please check the following point-by-point response. Our responses are highlighted.

Reviewer #1:

In their manuscript "Sequence and parent-of-origin dependent m6A contribute to allele-specific gene expression" Zhang et al. analyse m6A mRNA modifications and corresponding gene expression data to investigate allele-specific biases driven by sequence difference and/or parental origin. The authors posit that m6A modifications may contribute to allele-specific gene expression post-transcriptionally by driving allele-specific mRNA decay (and through other unspecified mechanisms).

In principle, this is an interesting and comprehensive analysis of a nice set of data, but as it stands, the paper is very difficult to read and understand. The results and figures can only be understood by reading the methods section in parallel and very carefully. The authors need to define much more explicitly what is being compared in their different analyses. Also, the figures are very busy; more is not always better. Some panels show the same data just in a slightly different way. It would make the paper easier to digest if the figures were slimmed down.

Response: Thank you for the careful evaluation of our manuscript and the helpful suggestions for improvement. We have taken these comments very seriously and have revised the manuscript substantially to improve clarity and readability.

1. *For any future revision, please do not separate figure legends from the figures and add line numbers. Here I will refer to page numbers and figures for detailed comments and questions.*

Response: Thank you for this suggestion. We have integrated figures and corresponding legends to be adjacent to each other. We have also added continuous line numbers throughout the manuscript. We hope these formatting changes will significantly improve the readability and review process.

2. *Page 2 Abstract: The abstract needs to mention what m6A is. It doesn't say that it's an mRNA modification.*

Response: Thank you for pointing out that the abstract lacked a definition of m⁶A. We have revised the abstract to include a description of m⁶A as the most prevalent post-transcriptional mRNA modification and highlighted its regulatory role in gene expression (page 2, lines 29-31).

This change, along with other revisions, ensures the abstract is both informative and adheres to the EMBO Journal's word limit of 175 words.

3. *Fig 1a: The diagram does not help to understand what the samples are. It would be better to have supplementary table 1 as a table in the main part of the paper. Especially the middle table in the diagram is very confusing. It's not clear how you get seven and nine samples from what looks like 2x4 age- and sex-matched offspring from reciprocal crosses.*

Response: Thank you for pointing out the ambiguity in Fig. 1a. We have removed the confusing middle table (Fig. 1A) and incorporated the details previously presented in Supplementary Table 1 as Table 1 in the main text (page 30, Table 1). This provides a clearer explanation of our samples.

4. *Fig 1d: Here it would help to connect, for example, the two sub-samples for one of the P0 male samples with a dotted line and to explain in the text that these come from the same physical sample.*

Response: Thank you for your suggestion regarding Fig. 1d. We have modified the figure to include dotted lines connecting sub-samples derived from the same physical sample (Fig. 1D). This visual enhancement, along with an explanatory statement in the figure legend (page 21, lines 541-542), improves the figure's clarity. Similar changes were also applied to Fig. EV1A.

5. *Page 7: It's confusing to call a pair of samples 'group' because group implies a number higher than 2. Make it very explicit in the text, e.g. say "The two age- and sex-matched F1i and F1r samples are from here on referred to as a group".*

Response: We appreciate your comment regarding the use of "group". Following your suggestion, we have added the clarification (page 7, lines 183-185). We hope this explicit definition will prevent any potential confusion regarding our use of the term 'group'.

6. *Page 8: If the A in the positive control m6As is a SNP how can the bias only be 80%? If this is a SNP, by definition the other allele has a base other than an A so can't have m6A.*

Response: Thank you for raising this important question. The 80% bias (instead of the expected 100%) in the positive control m⁶A sites reflects limitations of m⁶A-seq in m⁶A quantification, which relies on the immunoprecipitation of m⁶A-modified RNA fragments. A single fragment can cover multiple m⁶A sites. Therefore, clustered or adjacent m⁶A sites can interfere with each other's quantification. The presence of m⁶A sites near the positive control loci might influence the quantification of m⁶A methylation on the non-A allele, leading to observed levels above zero or even higher than the A-allele, despite the theoretical expectation of zero methylation. Furthermore, inherent non-specific antibody binding also introduces noise, which may also impact the allelic m⁶A quantification on the non-A allele (McIntyre et al, 2020).

However, the impact of these noise sources on our allele-specific analysis is negligible. Although approximately 20% of positive control m⁶A sites showed unexpected allelic bias (Fig. EV2B, C),

these sites exhibited very low allelic differences (upper quartile of $|\log_2(cpFC)| < 0.6$), which do not meet our stringent criteria for identifying A⁶Sm sites. To minimize noise, our analysis prioritized inter-sample reproducibility. This involved rigorous filtering of m⁶A sites, selection of significant A⁶Sm candidates ($P < 0.05$, FDR < 0.1 and $|\log_2(cpFC)| > 0.6$) present in both F1i and F1r samples within each F1i-F1r group, and restriction of all downstream analyses to highly reproducible A⁶Sm sites (detectable in at least four samples from the same tissue, with directional agreement between group- and tissue-level $\log_2(cpFC)$ or $\log_2(mpFC)$ values) (see Methods). The identified A⁶Sm sites are highly reproducible across F1i-F1r groups (Fig. 3A, B).

7. Fig 2a, b: I don't think these need to be in the main figure. Also, please define all abbreviations and labels in the figure legends. I had to dig through the methods to find what Ni+ etc are.

Response: We appreciate your comments on Fig. 2a, b. Following your suggestion, we have moved these figures to Fig. EV2 (Fig. EV2B, C). All figures in Fig. 2 and Fig. EV2 are now renumbered according to their order of appearance in the manuscript, and the figure legends have been revised to include definitions of all abbreviations and labels (see legends for Fig. EV2B, C).

8. Fig. 2e. The legend for the F1i-F1r group needs to be included here.

Response: Thank you for pointing out the missing legend. The legend for the F1i-F1r group has been included in the Figure (now Fig. 2C).

9. Page 10: Here the sentence "We then systematically assessed A⁶Sm across F1i-F1r groups" means A⁶Sm analysis WITHIN groups. On page 11 the authors write practically the same, i.e. "we next asked whether (...) seq-m⁶As show consistency across F1i-F1r groups", to mean BETWEEN groups. Given the complexity of this analysis it is paramount that the authors are very clear on what they mean.

Response: We appreciate your comments regarding the ambiguity in our description of the A⁶Sm analysis. We have addressed this by revising the text as follows.

On page 11 (now page 10), the analysis is a quantitative correlation analysis of allelic bias between F1i-F1r groups. To improve clarity, we replaced "across" with "among" (page 10, lines 251).

On page 10 (now page 9), the process incorporates both group- and tissue-level analyses, thus the original phrasing "across F1i-F1r groups" has been removed. The sentence is rewritten as: We then identified highly reproducible A⁶Sm based on their reproducibility in allelic bias direction (page 9, lines 230-231). The criteria for identifying highly reproducible A⁶Sm are detailed in the Methods section (page 42, lines 969-973; page 38, lines 873-878).

10. Page 10: '69% of testable sites exhibited group-sharing, which decreased to 41% for seq-m⁶As'. What does this mean? How are the testable sites different? Is the rest parent-driven?

Response: We apologize for the lack of clarity in our description of group-sharing. "Testable sites" for each F1i-F1r group were defined by stringent criteria applied to both F1i and F1r samples, requiring high allelic read counts (see Methods). Of all testable sites in our dataset, 69% showed consistent allelic bias across multiple F1i-F1r groups. The group-sharing percentage dropped to 41% when considering only the seq-ASm⁶As. The difference does not indicate that the remaining sites are parent-driven but rather reflects lower reproducibility among the identified seq-ASm⁶A sites meeting our stringent criteria. Our analysis revealed that most group-specific seq-ASm⁶As maintain consistent allelic bias directions in other groups where they were testable (but not identified as seq-ASm⁶A sites).

The relevant sections of the manuscript have been revised to improve clarity and remove ambiguity (page 10, lines 251-258).

11. *Fig 3g: It's not clear from the heatmap in what order the motifs are ranked. Also, setting NA values to zero is misleading. A better option would be to have a separate colour (a light grey) for the NA values.*

Response: Thank you for your helpful comments. We have clarified in the legend that motifs are ranked by occurrence frequency within all high-confidence m⁶A sites in each tissue (see the legends of Fig. 3F and Fig. EV3F) and now represent NA values in the heatmap as gray (Fig. 3F; Fig. EV3F).

12. *Fig 3h doesn't add much, it's just another depiction of the heatmap results. It's also not explained in the figure legend what the red numbers along the x-axis are.*

Response: We appreciate your comments regarding Fig. 3h (now Fig. 3G). While it does present data also shown in Fig. 3F, we consider the additional information on data distribution (rather than only mean values) is valuable for interpreting the heatmap. To facilitate a more comprehensive understanding, the figure illustrates the data distribution for two representative rows from the heatmap. As suggested, we have now updated the figure legend to clarify that the red numbers along the x-axis represent the number of m⁶A sites for each motif variation (page 25, line 587-588; Fig. 3G).

13. *Fig 3i: That correlation doesn't look great, weak R2. The data fall mostly into four distinct clusters along the x-axis.*

Response: We apologize for the lack of accuracy in our description of the figures (Fig. 3H; Fig. EV3G). We agree that the linear correlation is not strong. The figure primarily aims to illustrate a strong directional correlation between allelic m⁶A bias and motif variation effects (predicted by motifbreakR; see Methods). The majority of data points (quadrants II and IV) support the directional correlation. Relevant description has been updated to reflect this (page 11-12, lines 298-302).

14. *Page 15: "SNP effects on m6A consensus motif". From the text and figure panel it's not at all*

clear what is meant by that. Only after reading the methods can this be understood.

Response: We appreciate your comment regarding the unclear description of "SNP effects on m⁶A consensus motif." To improve clarity, we have revised this terminology to "motif variation effects (quantified by motifbreakR)" in the relevant texts and figures (page 11-12, lines 298-302; page 25, lines 590-592; page 44, line 1023; Fig. 3H; Fig. EV3G). Furthermore, a detailed explanation of the operation and underlying principles of motifbreakR has been provided in the Methods section (page 44, lines 1023-1033; see the 'Quantification of motif variation effects on m⁶A methylation' subsection).

15. Figure 4e: What are the numbers along the x-axis? Number of seq-ASm6A sites?

Response: Thank you for pointing out the lack of clarity in Figure 4e (now Fig. 4E). The numbers along the x-axis represent the count of seq-ASm⁶A sites. The figure legend has been updated to clarify this (page 26, lines 607-608).

16. Page 19: Here the wording needs to be made much clearer. "allelic biases in methylation (...) displayed a trend of favoring the parental allele with lower expression level" is very confusing. "Higher m6A levels correlated with lower gene expression levels" is probably meant here. Similarly "changes in allelic bias from m6A expression to methylation." What does this mean? Or "shifts in allelic bias towards the opposing allele" implies a measurement before and after methylation which is not the case. Overall this paragraph needs more precise phrasing.

Response: Thank you for your careful reading and helpful suggestions. We have revised the problematic sentences in this paragraph to improve clarity and accuracy. The specific changes are detailed below.

Original: Notably, the allelic biases in methylation reflected by $ARR_{mat,ip}$ values displayed a trend of favoring the parental allele with lower expression level (Fig. 5a).

Revised: Notably, we found that higher allelic m⁶A levels correlated to lower allelic expression levels, in a parent-of-origin dependent manner (Fig. 5A). (page 14, lines 369-371)

Original: We then employed the difference between $ARR_{mat,ip}$ and $ARR_{mat,input}$ to quantify changes in allelic bias from m⁶A expression to methylation. Remarkably, this analysis revealed widespread shifts in allelic bias towards the opposing parental allele in m⁶A methylation (Fig. 5b). These findings demonstrate the presence of parent-of-origin effects on allelic m⁶A methylation, characterized by a tendency towards preferential methylation of the parental allele opposite to the one favored in expression.

Revised: We then quantified the difference in parent-of-origin-dependent allelic bias between methylation and expression (calculated as $ARR_{mat,ip} - ARR_{mat,input}$) at these sites. A widespread bias for the opposing parental allele in m⁶A methylation compared to expression were observed (Fig. EV5C). These findings demonstrate a strong, parent-of-origin-dependent inverse correlation

between allelic m⁶A methylation and expression levels. (page 14, lines 375-380)

We have also revised other similar instances of potentially misleading phrasing throughout the manuscript to ensure greater precision and clarity:

Original: We then calculated the difference between $ARR_{c57,input}$ and $ARR_{c57,ip}$ to assess changes in allelic bias from m⁶A expression to methylation. Remarkably, widespread shifts in allelic bias towards the opposing allele in m⁶A methylation were observed (Fig. 4b; Extended Data Fig. 4c).

Revised: We then quantified the difference in sequence-dependent allelic bias between m⁶A expression and methylation (calculated as $ARR_{c57,ip} - ARR_{c57,input}$). This analysis revealed a remarkably widespread bias for the opposing allele in m⁶A methylation compared to expression (Fig. 4B; Fig. EV4C). (page 12, lines 321-324)

17. Fig 5b: Not sure what this adds beyond 5a.

Response: Figure 5a (now Fig. 5A) presents the overall distribution of maternal read ratios and the differences in these ratios between paired input and IP samples. Figure 5b (now Fig. EV5C) quantifies and displays the distribution of the ratio differences at each individual m⁶A site, providing a higher-resolution view not apparent in Fig. 5A. To improve conciseness, the original Figure 5b has been moved to Fig. EV5C.

18. Fig 5c. Are the dotted lines cut-offs for allele bias? Make clear in figure legend.

Response: Thank you for raising this question. Yes, the dotted lines in Fig. 5c (now Fig. 5B) represent the cutoffs for identifying parent-of-origin-dependent ASE. This has been clarified in the updated figure legend (page 28, lines 618-620). Similarly, the legend for Fig. EV5D was also revised.

19. Fig 5e. If ASE is present, one would expect the whole gene to be biased in the same direction. So not sure what is gained by looking at individual m6A sites across the gene.

Response: Thank you for raising this important question. Although expression of all m⁶A sites within a gene typically shows a consistent allelic bias direction, allelic methylation biases can vary among these sites. Gene-level analysis may not fully capture the complexities of m⁶A regulation. Studies have shown that individual m⁶A sites within the same gene can have distinct effects on gene expression, which are often determined by the specific m⁶A reader proteins involved (Murakami and Jaffrey, 2022). Some m⁶A sites promote mRNA stability (Huang et al, 2018), while others repress expression (Wang et al, 2014; Shi et al, 2017). This site-specific functional variability can be reflected in allelic bias patterns: some m⁶A sites show concordant allelic preferences for expression and methylation, while others exhibit opposing biases. To capture the site-specific patterns, we performed allele-specific analyses on individual m⁶A sites. Figure 5D (formerly Fig. 5e) shows the correlations between allelic expression and methylation biases at the

individual site level.

20. Page 21: "opposing" rather than "contradictory"

Response: Thank you for this suggestion. We have replaced "contradictory" with the more appropriate term "opposing" (page 15, lines 407). We have also corrected another similar instance (page 13, lines 334).

21. Page 24: "we quantified the allelic imbalance shift from expression to methylation". What does this mean? As above, "shift" implies a before and after, and a shift from expression to methylation doesn't make sense.

Response: Thank you for pointing this out. We agree that "shift" is inaccurate in this context. We have removed this sentence and have also corrected similar instances throughout the manuscript. The specific changes are detailed below.

Original: Remarkably, this analysis revealed widespread shifts in allelic bias towards the opposing parental allele in m⁶A methylation (Fig. 5b).

Revised: A widespread bias for the opposing parental allele in m⁶A methylation compared to expression were observed (Fig. EV5C). (page 14, lines 377-378)

Original: Remarkably, widespread shifts in allelic bias towards the opposing allele in m⁶A methylation were observed (Fig. 4b; Extended Data Fig. 4c).

Revised: This analysis revealed a remarkably widespread bias for the opposing allele in m⁶A methylation compared to expression (Fig. 4B; Fig. EV4C). (page 12, lines 323-324)

22. The authors state they propose two mechanisms, but the second is just a vague allusion to m⁶A reader binding. A bit more detail would be good.

Response: Thank you for this suggestion. We agree that the initial description of our second proposed mechanism, allele-specific m6A reader binding, was too vague. We have now revised the Discussion section to include additional details (pages 18-19, lines 480-497).

Reviewer #2

In this work the authors have mapped thousands of allele-specific m6A (ASm6A) sites with statistically significant. They show that over 90% of these sites are not due to motif sequence variations but most likely cis sequence changes within 50 nt sliding window. These allele-specific m6A sites mostly affect transcript decay, consistent with the known roles of m6A. The overall study design and execution are good, with valuable information provided. I think this is a good

contribution to the field and the results may stimulate future investigations. I have a few comments below:

Response: We are pleased that the Reviewer finds our study design, execution, and findings valuable.

1. Among thousands of sites identified if they used GLORI, how do they control potential deamination variations in GLORI. Is that a problem for their work?

Response: Thank you for raising this concern. We agree that potential deamination variations in the GLORI method could affect both qualitative and quantitative analyses of m⁶A sites.

To control potential deamination variations, we followed the library preparation strategies detailed in the GLORI protocol (Liu et al, 2023) and optimized data analysis methods. To minimize the impact of deamination variations on qualitative accuracy, we applied stringent criteria to define GLORI-detected m⁶A sites: 1) methylation rate > 0.1, and 2) read count > 8. High-confidence m⁶A sites for each m⁶A-seq sample were then selected by overlapping m⁶A peaks in that sample with GLORI-detected m⁶A sites in the corresponding tissue. Importantly, we avoided using GLORI data for quantitative allele-specific m⁶A analysis because potential deamination variations could confound SNP sites, potentially leading to false positives in determining the allelic origins of reads.

Using all these control processes, we believe that deamination variations in GLORI are not a problem in our work.

2. Could the authors speculate what are cis-regulatory factors for allele-specific m6A sites beyond the motif sequence? I assume many of these are allele-specific sequence changes that lead to altered RBP binding? Any specific motifs coming out of their analyses? It is hard to think all effects from these RBP binding to the target transcript. Whatelse?

Response: We appreciate these insightful questions regarding the *cis*-regulatory elements regulating allele-specific m⁶A beyond the known motif. A primary mechanism underlying ASm⁶A is likely the differential binding affinity of certain RBPs between alleles due to sequence variations. This differential binding could lead to differences in m⁶A modification between alleles. This is consistent with previous human m⁶A QTL studies which strongly suggest that sequence variation plays a critical role in regulating m⁶A levels via its effects on RBP binding (Zhang et al, 2020; Xiong et al, 2021).

To explore the relationship between RBP binding and ASm⁶A, we have employed two strategies: 1) motif enrichment analysis of sequences around SNPs in the flanking 50-nt region of ASm⁶A sites; and 2) identification of RBPs exhibiting preferential binding to ASm⁶A regions. The motif enrichment analysis (excluding sequences overlapping m⁶A motif positions) did not reveal any significantly enriched RBP motifs. This may indicate that allelic m⁶A regulation involves a heterogeneous collection of RBPs, with diverse binding specificities, making the identification of a common motif challenging. In the second analysis, using the binding sites of 47 RBPs curated by

the POSTAR2 database (Zhu et al, 2019), we identified RBPs with binding sites enriched near ASm⁶A regions compared to non-ASm⁶A controls in our dataset (see Response Figure 1 below). Due to the limited number of candidates and the lack of experimental validation, these findings were not included in the manuscript.

Additionally, we propose that RNA secondary structure is a potential regulator of ASm⁶A. Variations in RNA structure between alleles might affect the deposition of m⁶A. This hypothesis is supported by a study demonstrating a correlation between sequence variations impacting mRNA secondary structure and allelic m⁶A levels (Shachar et al, 2024).

Response Figure 1. Identification of candidate RBPs exhibiting preferential binding to seq-ASm⁶A regions in the cerebellum (left) and cerebrum (right). In the enrichment analyses, we compared RBP binding site numbers in the 100-nt flanking regions of seq-ASm⁶A and non-seq-ASm⁶A sites. The red dashed line represents the $P < 0.05$ threshold (Fisher's exact test). The blue dashed line represents the OR threshold. OR, odds ratio.

3. What function categories that may enrich parent-of-origin effect of allele-specific m⁶A sites?

Response: Thank you for raising this important question. Investigating the functional enrichment of genes with parent-of-origin-dependent ASm⁶A is crucial. However, the limited number of high-confidence parent-ASm⁶A sites (Fig. EV2F) currently prevents a robust functional enrichment analysis (e.g., Gene Ontology analysis). Nevertheless, our observation of pervasive parent-of-origin effects on m⁶A sites within imprinted genes (Fig. 5C, D) suggests a potential link to imprinted gene function. In mammals, imprinted genes influence a wide range of biological processes, including fetal growth, brain function, postnatal survival (e.g., thermoregulation, metabolism, feeding behavior), maternal care, sleep, stem cell maintenance, and are implicated in various diseases such as intrauterine growth restriction, obesity, diabetes, psychiatric disorders, and cancer (Tucci et al, 2019; Peters et al, 2014; Barlow et al, 2011).

We also observed highly reproducible parent-of-origin effects on m⁶A sites in known imprinted

genes, such as *Begain* and *Cobl* (Fig. 5D). *Begain*, a component of post-synaptic density protein complexes, contributes to learning and memory (Katano et al, 2023). *Cobl* is involved in actin filament organization (Beer et al, 2020) and regulation of neuronal morphology (Ahuja et al, 2007).

In summary, parent-of-origin effects on m⁶A methylation may influence a wide range of biological processes associated with imprinted genes. Further investigation with larger datasets and additional tissue types will be necessary to explore this further.

Reviewer #3:

This was a study of the post-transcriptional mRNA modification N6-methyladenosine (m6A) in two brain tissues of F1 hybrids between two divergent mouse strains. The main experiment used m6A-seq on 14 samples, using a method which relies on immunoprecipitation of fragmented barcoded pooled mRNA with anti m6A antibody. This results in sequencing only those mRNA fragments that contained m6A, and these were then also compared with non-immunoprecipitated fragmented barcoded pooled mRNA (the "input sample") to assess the overall abundance of transcripts independent of m6A. The second experiment used glyoxal and nitrate-mediated deamination (GLORI) treatment on two samples. This treatment deaminates only non-methylated adenosines to inosines, which can then be identified by comparing the resulting RNA-seq with reference genomes. By overlapping the datasets from these two different methods, high confidence m6A sites could be identified. The rest of the manuscript concerned an in depth bioinformatic analysis of these datasets to detect tissue, sex, age effects on m6A and strain and parent of origin effects on the allelic methylation and expression of transcripts as well as identifying cis-regulatory variants enriched in regions flanking m6A sites, making use of reciprocal crosses and paired sample design.

The major findings of the study were the identification of cis-regulatory variants in the 50 nucleotides to the 5' and 3' of m6A sites, and the detection of sequence dependent, and parent of origin dependent allele specific transcript methylation. It was particularly striking that there was a negative relationship between the allele specific methylation and allele specific expression, hinting at the possible role of allele-specific m6A in downregulating transcript abundance.

In mammalian cells m6A is the most abundant post-transcriptional mRNA modification and has been explored in many previous studies. The novelty in this study is the comparison of allele-specific expression with allele-specific methylation (adding the more robust method detecting m6A with both the competitive immunoprecipitation on pooled barcoded mRNA and cross comparison with GLORI seq), as well as the elucidation of sequence-dependent and parent of origin dependent effects. These novel finding presented in this manuscript therefore represent an important advance and will be of general interest and provide useful starting points for future research on the post transcriptional determinants of allele-specific expression. I do however have some reservations regarding the study.

Response: We are pleased that the Reviewer finds aspects of our study novel and impactful.

My major reservation is that although an in-depth study and analysis of m6A has been made for the F1 hybrids, the study provides no experiment or information on m6A in the equivalent tissues (cerebrum and cerebellum) and ages (0 and 7 days after birth) of the two parental mouse strains, C57 and PWK. Are they the same sites as seen in the C57 and PWK alleles in the F1? Without any information on the situation or different phenotypes of the parental strains it is hard to understand what is the biological relevance of the allele specific m6A and ASE in the F1 hybrids. On this theme one key study on transcriptome-wide m6A not cited or discussed is Xu et al 2021 <https://doi.org/10.3389/fpls.2021.685189>, which studied transcriptome-wide m6A in Arabidopsis strains and their hybrids. In this study Xu et al observed a disappearance of m6A differences at allelic sites in hybrids, with allelic bias in m6A in hybrids being very rare. This seems like a potentially large biological difference between the use of m6A in plants and animals that needs further investigation.

Response: We appreciate these insightful comments. We agree that data from the two parental mouse strains would be valuable for fully understanding the biological relevance of the ASm⁶A and ASE. Regarding the question of whether the m⁶A sites in the parental strains are identical to those in the F1 hybrids, we acknowledge that this cannot be definitively answered without data from parental strains. This is because m⁶A deposition is influenced not only by sequence motifs but also by cellular context. The cellular context may differ between parental strains and F1 hybrids, even within the same tissue at the same developmental stage, due to differing genetic backgrounds. Data from the parental strains would therefore be ideal for addressing this question.

However, data from parental strains are not always essential for allele-specific studies, particularly those aimed at identifying parent-of-origin and sequence-dependent effects on allele-specific events, and to explore the related molecular mechanisms (Xie et al, 2012; Perez et al, 2015; Gregg et al, 2010). Similarly, our study focused on identifying parent-of-origin and sequence-dependent effects on m⁶A methylation, predicting potential *cis*-regulatory elements, and exploring the correlation between ASm⁶A and ASE. Although these findings do not fully explain phenotypic regulation, they provide valuable insights into the molecular mechanisms underlying ASm⁶A regulation.

We have added a detailed discussion of the biological relevance of our findings in the Discussion section (page 19, lines 498-513). This discussion integrates existing knowledge of gene expression and m⁶A methylation in the mouse cerebrum and cerebellum.

We agree that future research should include a comparative analysis of m⁶A patterns in the parental strains and their F1 offspring. This comparison will be critical for determining whether the observed allele-specific m⁶A patterns in the F1 hybrids reflect inheritance from the parental strains or *de novo* establishment, and for assessing the biological significance of ASm⁶A.

Thank you for pointing out the omission of the key study about allelic m⁶A analysis in *Arabidopsis* and for highlighting the potential significance of their findings on the lack of allelic m⁶A bias in plant

hybrids. We have now cited and discussed this study in the Discussion section, emphasizing the intriguing contrast between plant and mammalian m⁶A regulation, and suggesting this as an important area for future research (page 19, lines 513-518).

My other reservations are more minor. As line numbers do not appear in the manuscript I received I have used page numbers and section headings to refer to the locations in the manuscript described below.

1. On p39 of the Methods (section "ASm⁶A identification and classification") it is stated that a binomial model was employed to assess the statistical significance of allelic methylation imbalance. Was this read count data assessed for overdispersion, which is a very common feature of this type of data (Castel et al. Genome Biology (2015) 16:195 DOI 10.1186/s13059-015-0762-6). If the data was over dispersed then this could have resulted in an overinflation of the significance levels calculated using a binomial model. The data should be tested for distribution and if the data is indeed over dispersed, a beta binomial or some other more appropriate model should be used instead.

Response: We appreciate the insightful comment regarding the potential impact of overdispersion in allelic read count data on ASm⁶A identification. We fully agree that allele-specific expression (ASE) data can often exhibit overdispersion under a binomial distribution (null hypothesis: equal expression of both alleles, with allelic read ratio = 0.5). This is likely due to a combination of biological and technical factors that are difficult to definitively separate (Castel et al. 2015). We carefully considered this potential issue and took several measures to mitigate its impact on our ASm⁶A analysis and ensure the robustness of our findings.

Two well-established approaches can effectively address such overdispersion. One approach involves analyzing ASE across multiple individuals and tissues to control for confounding factors and isolate the biological signal of interest, such as *cis*-regulatory variation and parent-of-origin effects (Castel et al. 2015). Another approach simulated read counts with overdispersion levels estimated from genomic DNA sequencing data (to generate **the "null" allelic read counts and allelic read ratio for each SNP**), and subsequently calculated the posterior probability of ASE for each simulated gene (Skelly et al, 2011). This method effectively reduces the false positive rate in ASE identification resulting from overdispersion.

While we acknowledge that ideally, a beta-binomial or similar model accounting for overdispersion could be considered, there are differences between ASE analysis (based on RNA-seq data) and our ASm⁶A identification (based on m⁶A-seq data). In m⁶A-seq data, the untreated input data provide **the "null" allelic read counts and allelic read ratio for each m⁶A site**. Specifically, for each testable m⁶A site, we used the binomial test to assess whether the allelic read ratio (ARR) in the IP sample differed significantly from that in the paired untreated input sample. The null hypothesis for this binomial test is that allelic read ratios are equal in both IP and input samples. This approach inherently controls for baseline allelic imbalances in expression when testing for allele-specific methylation. Therefore, we chose a binomial test approach combined with rigorous mitigation strategies. We believe this effectively address the concern in the specific context of our

ASm⁶A identification. Our justification and control measures are as follows:

Our approach directly tests for allelic methylation bias (compared to allelic expression bias) at each site. Details are as follows:

$$ARR_{c57,input} = \frac{R_{c57,input}}{R_{c57,input} + R_{pwk,input}}$$

$$R_{c57,ip} \sim \text{Binomial}(n = R_{c57,ip} + R_{pwk,ip}, p = ARR_{c57,input}),$$

Here, $R_{c57,input}$ and $R_{pwk,input}$ represent the read counts in the untreated input sample for C57 and PWK alleles, respectively. $R_{c57,ip}$ and $R_{pwk,ip}$ represent the read counts in the paired IP sample for C57 and PWK alleles, respectively. $ARR_{c57,input}$ denote the ARR value obtained from the untreated input sample for the C57 allele.

To minimize the potential impact of overdispersion, we employed exceptionally stringent filtering criteria to select testable m⁶A sites for binomial testing: $R_{c57,input} \geq 5$, $R_{pwk,input} \geq 5$, $R_{c57,input} + R_{pwk,input} \geq 20$, and $R_{c57,ip} + R_{pwk,ip} \geq 30$. These criteria substantially reduce the influence of low read counts on our analysis.

In addition to the P and FDR values from the binomial test, we considered allelic methylation differences ($\log_2(cpFC)$) and inter-sample reproducibility for ASm⁶A identification. To ensure robustness, we applied stringent criteria: only significant ASm⁶A candidates ($P < 0.05$, FDR < 0.1 and $|\log_2(cpFC)| > 0.6$) present in both F1i and F1r samples were retained within each F1i-F1r group. Furthermore, all downstream analyses were based on highly reproducible ASm⁶A sites (detectable in at least four samples from the same tissue, with directional agreement between group- and tissue-level $\log_2(cpFC)$ or $\log_2(mpFC)$ values; see Methods for details).

Importantly, the high reproducibility of our identified ASm⁶A sites in allelic bias levels and directions across independent F1i-F1r groups (Fig. 3A, B) provides strong empirical evidence that our stringent approach has effectively controlled for potential noise and overdispersion, and that our ASm⁶A findings are robust and biologically meaningful.

In conclusion, while acknowledging the potential for overdispersion in read count data, we are confident that our binomial test design, combined with exceptionally stringent filtering and a focus on reproducibility, effectively addresses this issue. This ensures the reliability and robustness of our identified ASm⁶A sites and downstream results.

2. In the introduction and at multiple points throughout the results and discussion the mechanism of m6A exerting its function by promoting mRNA degradation is mentioned. On p16 in the results section it is mentioned that "widespread shifts in allelic bias towards the opposing allele in m6A methylation were observed" - which I take as meaning that the allele with higher overall expression tends to have lower m6A methylation. I find it very hard to assess on the data in the manuscript alone which is a measurement of the steady state expression levels and m6A methylation at a

snapshot in time whether this conclusion is justified in this context. This is because there is no attempt to quantify transcript degradation in this manuscript. The opposing directions of the expression and methylation signals do not imply causation - it could just be that linked SNPs that are responsible both for cis-regulation of expression and for methylation. To make such a certain statement of the mechanism would require testing this hypothesis by for example measuring transcript abundance over time while either blocking or not blocking m6A methylation, or employing gene editing techniques to swap m6A motif sites between the two strain genetic backgrounds. The proposal on p18 that "Based on these findings, we propose that seq-ASm6As primarily participate in regulating ASE by promoting allele-specific mRNA degradation" should therefore be less overstated.

Response: Thank you for your insightful comments. We agree that our current data do not definitively establish a causal link between allele-specific m⁶A methylation and expression. Our hypothesis is based on a combination of our observations and existing literature: numerous studies have demonstrated that the major function of m⁶A across the mammalian transcriptome is to promote mRNA degradation in the cytoplasm (Murakami and Jaffrey, 2022; Zaccara and Jaffrey, 2020; Lee et al, 2020; Shi et al, 2017; Wang et al, 2014). Since allele-specific m⁶A sites are a subset of transcriptome-wide m⁶A, it is logical that they share this dominant function.

We agree that, lacking direct experimental validation, claiming ASm⁶As primarily regulate ASE by promoting allele-specific mRNA degradation is an overstatement. Specifically, we have revised our language throughout the manuscript to emphasize a potential regulatory role for ASm⁶A in negatively regulating ASE, rather than definitively stating that allele-specific mRNA degradation is the primary mechanism (page 6, lines 136-137; page 13, lines 346-349; pages 16-17, lines 439-441; page 18, 483-493). We now consistently use more cautious phrasing, such as 'suggesting a potential role', 'potentially through mRNA degradation', or 'consistent with the well-established function of m⁶A' instead of more assertive statements about primary mechanisms or causation. As suggested, we have also added a discussion of the limitations of our analysis and the need for future research to confirm this mechanism by measuring allelic transcript abundance over time while either blocking or not blocking allele-specific m⁶A methylation (page 18, 491-493).

3. I spent a lot of time wondering why were cerebrum and cerebellum selected, and why postnatal day 0 and day 7? The explanation for the choice of tissue is hidden away in the methods section (under subheading "Multiplexed m6A-seq), and there is no explanation for the choice of ages. This should be included in the introduction or results.

Response: Thank you for pointing out the lack of explanation regarding the selection of tissues and ages. This information has now been included in the Introduction section (page 5, lines 120-126), outlining the rationale for selecting the cerebrum and cerebellum tissues at postnatal days 0 (P0) and 7 (P7). We chose to study the cerebrum and cerebellum because both tissues are known to play critical roles in brain development and function, and both allele-specific expression (ASE) and m⁶A methylation are recognized as important regulatory mechanisms in these processes. We selected P0 and P7 because these represent critical early postnatal stages of brain development, where m⁶A methylation is known to be dynamically regulated, suggesting a key role

in early brain development.

4. *Frequent mention of age matched pairs of samples from the reciprocal crosses was made, but to make sense of the pairs it was necessary to look very carefully at TableS1, as figure 1a was not easy to interpret. Although the total number of samples sequenced was 16, there was only one matched pair for each combination of conditions of sex, age and tissue.*

Response: Thank you for pointing out the difficulty in interpreting Figure 1a. We have removed the confusing middle table (Fig. 1A) and incorporated the details previously presented in Supplementary Table 1 as Table 1 in the main text (page 30). This provides a clearer explanation of our samples.

Although only one F1i-F1r pair was available for each combination of sex, age, and tissue, the reciprocal crosses design provides biological replicates in our allele-specific analysis for each combination. Thus, the F1i and F1r samples within each age- and sex-matched pair serve as biological replicates for assessing allele-specific m⁶A methylation while controlling for parental origin. The frequent mention of age- and sex-matched pairs highlights our efforts to minimize the influence of developmental stage and sex on allele-specific analysis and ASm⁶A identification within each F1i-F1r group.

5. *IP (m6A immunoprecipitation) is first mentioned in manuscript the results section on p16 in this sentence "To address this, we calculated paired allelic read ratios (ARRs) for both input (expression) and IP (methylation) samples at each m6A site, $ARR_{c57,input}$ and $ARR_{c57,ip}$, quantifying sequence-dependent allelic preferences (see Methods)." The explanation of what "IP" stands for is not given until the methods section on p31: "Multiplexed m6A immunoprecipitation was conducted 2 rounds, based on published work". The multiplexing is an important part of this protocol to make the m6A detection comparable between samples and more explanation of this is needed in the methods, not requiring all readers to go and read Dierks et al 2021 methods paper in full. The explanation for what IP/IP samples mean should also be brought forwards to the results section. I found this particularly confusing as the IP samples were compared with input samples, which also sounds a bit similar to IP. This would be less confusing if the more meaningful label of "expression samples" and "methylation samples" as used on page 16 was used throughout, or adding "untreated" before "input".*

Response: Thank you for your suggestion. We have included the full term for IP at its first mention (page 12, lines 314), and added more details about multiplexed m⁶A immunoprecipitation in the Methods section (page 32-33, lines 706-715). For improved clarity, we have also added "untreated" before each mention of "input" samples throughout the manuscript.

6. *P0 and P7 are not explained until the methods section and yet are mentioned in the results and figures. Non mouse biologists will not be familiar with these terms. Explain them at the first mention.*

Response: Thank you for pointing this out. As suggested, we have added definitions for P0 and

P7 where they first appear in the manuscript (page 5, line 122).

7. Figures: A lot of the figures were unnecessarily large and complicated to read. This was compounded by the choice of extremely similar colours, for example dark blue and teal green in Figure 1d. Some panels e.g. 1e, 2e, are showing very similar trends across the samples and could be replaced with either the average or an example and the explanation that the other samples followed the same trend. 3b did not require a plot and could be replaced with a single sentence in the text. 5c has no explanation for the ordering on the vertical axis - I presume the genes are ranked by maternal read ratio? Supplementary figure 2a there no need for a discontinuity in the graph, simply make the plot taller, it's supplementary so space constraint not an issue. Also in supplementary figure 2a the groups of m⁶A sites are in almost indistinguishable shades of blue. Figure 5f had an excessive number of symbols and colours in the key making it virtually unreadable. It would have been as easy to just label the data points with text.

Response: We greatly appreciate your detailed reading of the figures and your helpful suggestions for improving their clarity and readability. In response to your feedback, we have carefully revised the figure presentations as follows:

In Figures 1d (now Figure 1D) and Figure EV1G, we have replaced dark blue and teal green with more contrasting colors to ensure clear differentiation between C57 and PWK strains.

As suggested, Figure 1e (Fig. 1E) now shows a representative example with the legend explaining that other samples exhibit similar trends.

While we acknowledge that Figure 2e (now Fig. 2C) shows similar trends across groups, we have chosen to retain this figure because it also presents valuable information on the number and distribution of highly reproducible ASm⁶A sites in each F1i-F1r group, revealing variations among these groups.

As suggested, we have removed Figure 3d and replaced it with a textual description of the relevant observations (page 10, lines 268-270)

The legends for Figure 5c (now Fig. 5B) and Fig. EV5D have been revised to clarify that the genes are ranked by maternal read ratio (see the legends for Fig. 5B and Fig. EV5D).

We appreciate your helpful comments on Supplementary Figure 2a (now Figure EV2A). We have modified the color scheme to enhance clarity. The current y-axis scale has been retained to ensure clear visualization of the differences in the number of testable m⁶A sites across groups, which is crucial for understanding our reproducibility analysis.

For Figure 5f (now Fig. 5E), as suggested, we have replaced the symbols with text labels to clearly indicate the genomic location of each data point.

We believe these extensive figure revisions, implemented based on your detailed and insightful

feedback, have significantly enhanced the clarity, conciseness, and overall effectiveness of our figures in communicating our key findings.

8. Punctuation on the same line as equations in the methods section leads to confusion. There is unnecessary and confusing punctuation after every equation e.g. p34 comma after equation for $L_{ij} = \log\left(\frac{p_{ij}/t_{ij}}{p_j/T_j}\right)$, on p36 full stop after equation for $\log_2(cpFC) = \log_2\left(\frac{L_{cs7}}{L_{pwk}}\right)$. Something strange is going on with brackets on p38 in the methods section (mixture of normal and square). It was unclear to me if the bracket shapes have any significance.

Response: We appreciate your comments regarding the punctuation following equations. We have revised the manuscript to remove all punctuation immediately following equations for clarity.

The use of parentheses and square brackets on page 38 (now page 40, lines 913-915) to define ARR ranges denotes open and closed intervals, respectively. The relevant sentences are as follows: "Based on $ARR_{mat,input}$ values, we categorized allelically detectable m⁶A sites within imprinted genes into four groups: (0.6,1] for strong maternal bias; (0.5,0.6] for mild maternal bias; [0.4,0.5) for mild paternal bias; [0,0.4) for strong paternal bias." The $ARR_{mat,input}$ was calculated as follows (see Methods):

$$ARR_{mat,input} = \frac{R_{mat,input}}{R_{mat,input} + R_{pat,input}}$$

The $ARR_{mat,input}$ value, ranging from 0 to 1, reflects the relative expression of maternal and paternal alleles. **A value of 0 indicates exclusive paternal allele expression (maternal allele silencing), while a value of 1 indicates exclusive maternal allele expression (paternal allele silencing).** We divided the entire range [0, 1] into four categories of allelic bias: (0.6, 1] for strong maternal bias; (0.5, 0.6] for mild maternal bias; [0.4, 0.5) for mild paternal bias; [0, 0.4) for strong paternal bias. The endpoints representing the most extreme biases (0 and 1) are included in the respective categories using square brackets (closed intervals). The midpoint, 0.5, representing balanced expression, is excluded and denoted using open intervals.

9. On p26 in the legend of Figure 1c it is unclear what the n=2153 referring to?

Response: Thank you for pointing out the ambiguity in the legend of Figure 1c (now Fig. 1C). We have updated the legend to specify that $n = 2,153$ refers to the total number of allelically detectable m⁶A sites shared across all samples (page 21, line 537).

10. On p37 in the methods section mention is made in the "Inter-sample same-strain and intra-sample different-strain allelic m6A imbalance to "negative control datasets". This was inadequately explained what it is controlling for - is this trying to work out the background level of allelic methylation?

Response: We apologize for the lack of clarity in our previous description. The "negative controls"

represent allelic m⁶A comparisons between alleles from the same strain, used to assess background allelic m⁶A imbalance level independent of strain-dependent effects. Specifically, for each m⁶A site, the inter-sample same-strain allelic m⁶A imbalance level served as a negative control, allowing us to identify significant m⁶A differences between alleles from different strains. Our analysis revealed significantly greater allelic m⁶A differences between strains than within strains, highlighting the strain- or sequence-dependent effects on allele-specific m⁶A profiles. For clarity, we have added further explanation to the Methods section (pages 38-39, lines 881-886).

11. *There was a bit of confusion between where 100 nucleotide and 50 nucleotide regions flanking m6A sites were being considered between the text and the figures. If I understood correctly, 100 nucleotide flanking regions were first considered but all identified potential cis-regulatory sites fell within the 50 nucleotide flanking regions? This should be made clearer in the text.*

Response: Thank you for pointing out the ambiguity. Our initial analysis involved the consideration of 100-nt flanking regions. However, all identified *cis*-regulatory elements were found to lie within the 50-nt regions. This has been clarified in the manuscript (page 11, lines 279-289).

Referee crosscommenting:

Reviewer 3 here. I just want to take this opportunity to further explain my comment regarding the lack of information on gene expression and m6A in the parental strains. My comment was that it is difficult to understand the biological significance of the allele-specific ASE and m6A in the F1 without information regarding m6A in the parental strains. I did not necessarily expect this to be addressed by the generation of the (large) missing datasets as I understand this would be time consuming if there is no data available. I think a good discussion on this point citing the literature regarding the two strains and discussing the relevance of cerebrum and cerebellum gene expression and posttranscriptional modifications to mouse biology would be sufficient. So perhaps we are not in as much disagreement as it seems.

Response: We are deeply grateful to the Reviewer for taking the time to further clarify their concerns regarding the biological significance of parental strain data. We sincerely appreciate this additional guidance, which is immensely helpful for refining our manuscript. We fully understand and agree that the absence of parental strain m⁶A data makes it more challenging to definitively interpret the biological relevance of A⁵m⁶A and ASE. We are particularly thankful that the Reviewer has clarified that generating these extensive datasets is not required for this revision, and that a more robust Discussion section, incorporating existing literature, will be sufficient to address their major reservation.

In direct response to the Reviewer's suggestions, we have significantly expanded the Discussion section (page 19, lines 498-518) to more thoroughly address the biological significance of our findings. This revision integrates existing knowledge on gene expression and m⁶A methylation in the mouse cerebellum and cerebrum, providing broader context for our results. We hope the revised manuscript now provides a much more robust and compelling account of the biological

significance of our findings.

References

- Ahuja R, Pinyol R, Reichenbach N, Custer L, Klingensmith J, Kessels MM, Qualmann B (2007) Cordon-bleu is an actin nucleation factor and controls neuronal morphology. *Cell* 131: 337-50
- Barlow DP (2011) Genomic imprinting: a mammalian epigenetic discovery model. *Annu Rev Genet* 45: 379-403
- Beer AJ, Gonzalez Delgado J, Steiniger F, Qualmann B, Kessels MM (2020) The actin nucleator Cobl organises the terminal web of enterocytes. *Sci Rep* 10: 11156
- Gregg C, Zhang J, Weissbourd B, Luo S, Schroth GP, Haig D, Dulac C (2010) High-resolution analysis of parent-of-origin allelic expression in the mouse brain. *Science* 329: 643-8
- Huang H, Weng H, Sun W, Qin X, Shi H, Wu H, Zhao BS, Mesquita A, Liu C, Yuan CL, Hu YC, Huttelmaier S, Skibbe JR, Su R, Deng X, Dong L, Sun M, Li C, Nachtergaele S, Wang Y et al. (2018) Recognition of RNA N(6)-methyladenosine by IGF2BP proteins enhances mRNA stability and translation. *Nat Cell Biol* 20: 285-295
- Katano T, Konno K, Takao K, Abe M, Yoshikawa A, Miyakawa T, Sakimura K, Watanabe M, Ito S, Kobayashi T (2023) Brain-enriched guanylate kinase-associated protein, a component of the post-synaptic density protein complexes, contributes to learning and memory. *Sci Rep* 13: 22027
- Lee Y, Choe J, Park OH, Kim YK (2020) Molecular Mechanisms Driving mRNA Degradation by m(6)A Modification. *Trends Genet* 36: 177-188
- Liu C, Sun H, Yi Y, Shen W, Li K, Xiao Y, Li F, Li Y, Hou Y, Lu B, Liu W, Meng H, Peng J, Yi C, Wang J (2023) Absolute quantification of single-base m(6)A methylation in the mammalian transcriptome using GLORI. *Nat Biotechnol* 41: 355-366
- McIntyre ABR, Gokhale NS, Cerchiatti L, Jaffrey SR, Horner SM, Mason CE (2020) Limits in the detection of m(6)A changes using MeRIP/m(6)A-seq. *Sci Rep* 10: 6590
- Murakami S, Jaffrey SR (2022) Hidden codes in mRNA: Control of gene expression by m(6)A. *Mol Cell* 82: 2236-2251
- Perez JD, Rubinstein ND, Fernandez DE, Santoro SW, Needleman LA, Ho-Shing O, Choi JJ, Zirlinger M, Chen SK, Liu JS, Dulac C (2015) Quantitative and functional interrogation of

parent-of-origin allelic expression biases in the brain. *Elife* 4: e07860

- Peters J (2014) The role of genomic imprinting in biology and disease: an expanding view. *Nat Rev Genet* 15: 517-30
- Shachar R, Dierks D, Garcia-Campos MA, Uzonyi A, Toth U, Rossmannith W, Schwartz S (2024) Dissecting the sequence and structural determinants guiding m6A deposition and evolution via inter- and intra-species hybrids. *Genome Biol* 25: 48
- Shi H, Wang X, Lu Z, Zhao BS, Ma H, Hsu PJ, Liu C, He C (2017) YTHDF3 facilitates translation and decay of N(6)-methyladenosine-modified RNA. *Cell Res* 27: 315-328
- Skelly DA, Johansson M, Madeoy J, Wakefield J, Akey JM (2011) A powerful and flexible statistical framework for testing hypotheses of allele-specific gene expression from RNA-seq data. *Genome Res* 21: 1728-37
- Tucci V, Isles AR, Kelsey G, Ferguson-Smith AC, Erice Imprinting G (2019) Genomic Imprinting and Physiological Processes in Mammals. *Cell* 176: 952-965
- Wang X, Lu Z, Gomez A, Hon GC, Yue Y, Han D, Fu Y, Parisien M, Dai Q, Jia G, Ren B, Pan T, He C (2014) N6-methyladenosine-dependent regulation of messenger RNA stability. *Nature* 505: 117-20
- Xie W, Barr CL, Kim A, Yue F, Lee AY, Eubanks J, Dempster EL, Ren B (2012) Base-resolution analyses of sequence and parent-of-origin dependent DNA methylation in the mouse genome. *Cell* 148: 816-31
- Xiong X, Hou L, Park YP, Molinie B, Consortium GT, Gregory RI, Kellis M (2021) Genetic drivers of m(6)A methylation in human brain, lung, heart and muscle. *Nat Genet* 53: 1156-1165
- Zaccara S, Jaffrey SR (2020) A Unified Model for the Function of YTHDF Proteins in Regulating m(6)A-Modified mRNA. *Cell* 181: 1582-1595 e18
- Zhu Y, Xu G, Yang YT, Xu Z, Chen X, Shi B, Xie D, Lu ZJ, Wang P (2019) POSTAR2: deciphering the post-transcriptional regulatory logics. *Nucleic Acids Res* 47: D203-D211

Dear Dr. Luo,

thank you for submitting a revised version of your manuscript. Your study has now been seen by two of the original referees, who find that their previous concerns have been addressed and now recommend publication of the manuscript. Referee #1 was not available for re-review but we think that you thoroughly addressed the concerns raised by this referee during the first round of revision. There remain only a few mainly editorial points that have to be addressed before I can extend formal acceptance of the manuscript:

- On the abstract page of the manuscript, please include 4-5 general keyword terms to enhance searchability.
 - Please adjust the format of the reference list and of the in-text citations according to EMBO Journal format (alphabetical order, author name et al + year.../up to 10 author names in the reference list before et al / please refer to our Guide to Authors for additional information on EMBO J reference format).
 - Please rename the Conflict of Interest section into "Disclosure and Competing Interests Statement", in accordance with our updated Guide to Authors (<https://www.embopress.org/competing-interests>)
 - As we are switching from a free-text author contribution statement towards a more formal statement based on Contributor Role Taxonomy (CRediT) terms, please remove the present Author Contribution section and instead specify each author's contribution(s) directly in the Author Information page of our submission system during upload of the final manuscript. See <https://casrai.org/credit/> for more information.
 - Please upload EV figures as individual Figure files and the include the legends in ms file
- Please update the DATASET EV LEGENDS: for Tables EV2 and EV3 source file names, titles, legends and manuscript callouts all need to be updated to Dataset EV1-EV2, legends should be uploaded as a separate tab/sheet in each Excel file
- Please convert the APPENDIX 1 FILE WITH ToC into PDF format;
 - Please provide suggestions for a short 'blurb' text prefacing and summing up the conceptual aspect of the study in two sentences (max. 250 characters), followed by 3-5 one-sentence 'bullet points' with brief factual statements of key results of the paper; they will form the basis of an editor-written 'Synopsis' accompanying the online version of the article. Please also provide an altered synopsis image, making sure that the aspect ratio conforms to our website's format - it should be exactly 550 pixels wide and between 300-600 pixels high.
 - Please provide the specific URL for GSE265979 dataset in the data availability statement.
 - Figure Legends (main + EV):
1. Please note that the exact p values are not provided in the legends of figures 1E, 2D, 3D, H; 4A, B, E; 5A; EV3 G, EV4 B, C; EV5 C
 2. Please indicate the statistical test used for data analysis in the legends of figures 2D, EV3 G.
 3. Please note that the box plots need to be defined in terms of minima, maxima in the legends of figures 2A, 3E, G; 4A, B, E; 5A, C, D, E; EV2 B-D; EV3 C, E; EV4 B, C; EV5 B, C, E.
 4. Please note that information related to n is missing in the legends of figures 2B, 3E, G; 5A, B, D; EV3 E, EV4 B, C; EV5 B-E.
- Please rename Table EV4 to Table EV2 with the corresponding callout
 - Section order should be corrected: Title page - Abstract & Keywords - Introduction - Results - Discussion - Methods - Data Availability - Acknowledgements - Disclosure and Competing Interests Statement - References - Figure Legends - Table(s) - Expanded View Figure Legends.

With best regards,

Cornelius Schneider

Cornelius Schneider, PhD
Editor | The EMBO Journal
c.schneider@embojournal.org

We realize that it is difficult to revise to a specific deadline. In the interest of protecting the conceptual advance provided by the work, we recommend a revision within 3 months (29th Jun 2025). Please discuss the revision progress ahead of this time with

the editor if you require more time to complete the revisions. Use the link below to submit your revision:

Referee #2:

The authors have addressed my comments.

Referee #3:

The authors have thoroughly addressed my comments on the original manuscript (EMBOJ-2024-118564) and in particular I am now satisfied with the additional discussion of the biological relevance of the findings in the discussion section.

I previously summarised the study and commented on the significance of the findings on my review of the original manuscript and I still stand by my original review on that (that it indeed both novel and significant and of general interest) so I have not repeated it here.

The clarity of the figures and tables and indeed the entire manuscript is also much improved, in line with the comments from myself and the other reviewers. I am therefore now happy to recommend that the manuscript is suitable for publication.

All editorial and formatting issues were resolved by the authors.

Dear Dr. Luo,

I am pleased to inform you that your manuscript has been accepted for publication in the EMBO Journal.

Yours sincerely,

Cornelius Schneider, PhD
Editor
The EMBO Journal
c.schneider@embojournal.org
